# Learning optimal policies through contact in differentiable simulation

## Abstract

Model-Free Reinforcement Learning (MFRL) has garnered significant attention for its effectiveness in continuous motor control tasks. However, its limitations become apparent in high-dimensional problems, often leading to suboptimal policies even with extensive training data. Conversely, First-Order Model-Based Reinforcement Learning (FO-MBRL) methods harnessing differentiable simulation offer more accurate gradients but are plagued by instability due to exploding gradients arising from the contact approximation model. We propose Adaptive Horizon Actor Critic (AHAC), a massively parallel FO-MBRL approach that truncates trajectory gradients upon encountering stiff contact, resulting in more stable and accurate gradients. We experimentally show this on a variety of simulated locomotion tasks, where our method achieves up to 64 % higher asymptotic episodic reward than state-of-the-art MFRL algorithms and less hyper-parameter sensitivity than prior FO-MBRL methods. Moreover, our method scales to high-dimensional motor control tasks while maintaining better wall-clock-time efficiency. `https://adaptive-horizon-actor-critic.github.io/`

## 1 Introduction

Reinforcement Learning (RL) has achieved remarkable success in complex tasks, such as Atari games (Mnih et al., 2013), Minecraft (Hafner et al., 2023) and Go (Silver et al., 2017). Combined with the Policy Gradients Theorem (Sutton et al., 1999), we can derive approaches for solving continuous motor control tasks. Although some of these Model-Free Reinforcement Learning (MFRL) approaches have achieved impressive results (Hwangbo et al., 2017; Akkaya et al., 2019; Hwangbo et al., 2019), they suffer from subpar sample efficiency, limiting their practical utility (Amos et al., 2021). Consequently, addressing this limitation has been a central research focus for the past few years.

An alternative approach, Model-Based Reinforcement Learning (MBRL), seeks to model the environment's dynamics for improved efficiency. However, most MBRL methods still rely on experience data to learn dynamics and often produce suboptimal policies compared to MFRL. To enhance sample efficiency, high-performance physics simulators can be employed, offering abundant training data in ideal scenarios. When combined with computationally efficient MFRL methods, these simulators enable quick training of robots for tasks such as walking (Rudin et al., 2022). However, a question remains: even with extensive data, can MFRL effectively tackle high-dimensional motor control problems?

Given the substantial effort put into producing accurate and efficient simulators, one should naturally ask, why don't we use them as models for MBRL? In turn, this makes it tempting to learn the policy using first-order methods which are theoretically more efficient (Berahas et al., 2022). This has been explored in the domain of model-based control literature, where we differentiate the model in order to plan trajectories for applications such as autonomous driving (Kabzan et al., 2019) and agile quadrotor maunders (Kaufmann et al., 2020). However, using first-order methods to learn feedback policies in standard RL settings has received limited attention. Non-differentiable contact point discontinuities in available simulation models have been a major hurdle, leading to the development of differentiable simulators (Hu et al., 2019a; Freeman et al., 2021; Heiden et al., 2021; Xu et al., 2021).

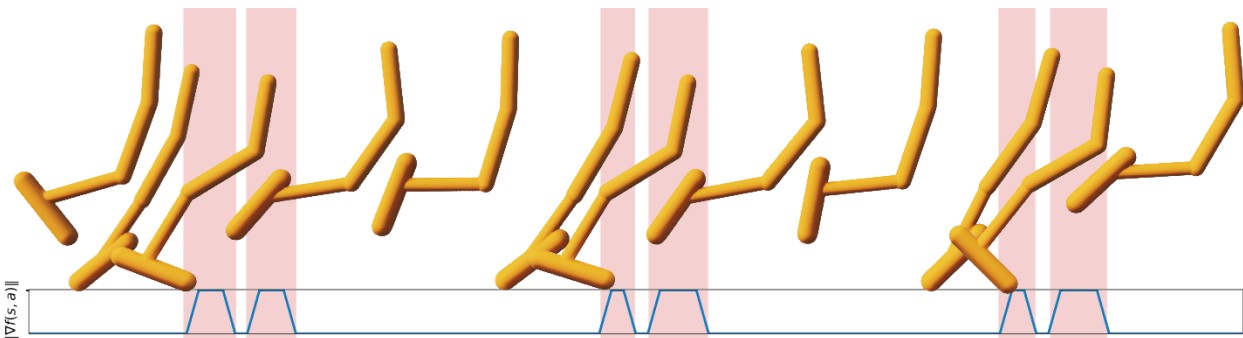

Figure 1: We find that FO-MBRL methods suffer from high dynamics gradients $\|\nabla f(s,a)\| \gg 0)$ which often arise from stiff contact approximation. Our proposed method, AHAC, truncates model-based trajectories at the point of and during stiff contact, thus avoiding both the gradient bias and learning instability exhibited by previous methods using differentiable simulation.

Where model-based control literature hand-designs bespoke models for each problem, differentiable simulation aims to create a physics engine that is fully differentiable. Thus, applying it to a different problem is as easy as defining the structure of the problem (e.g. joints and links) and leaving the physics to be calculated by the engine. These simulators have enabled a new family of FO-MBRL algorithms that can efficiently learn complex control tasks. Short Horizon Actor Critic (SHAC) (Xu et al., 2022) is such an approach that utilises the popular actor-critic paradigm (Konda & Tsitsiklis, 1999). The actor is trained in a first-order fashion, while the critic is trained model-free. This allows SHAC to learn through the highly non-convex landscape by using the critic as a smooth surrogate of the cumulative reward objective and avoiding exploding gradients by employing short-horizon rollouts. While SHAC boasts incredible sample efficiency when compared against MFRL, it is also brittle, exhibits higher learning instability, and requires extensive hyper-parameter tuning (Suh et al., 2022).

In this study, we attempt to address those issues and shift our focus from sample efficiency to the asymptotic performance of FO-MBRL methods in massively parallel differentiable simulations. We aim to answer the following questions:

1. What causes learning instability in FO-MBRL approaches such as SHAC? Our analysis reveals that first-order methods exhibit high empirical bias when estimating gradients through sample-limited Monte-Carlo approximation, hindering efficiency and resulting in suboptimal policies. This bias is primarily driven by the high magnitude dynamical gradients ($\|\nabla f(\boldsymbol{s}, \boldsymbol{a})\| \gg 0$) arising from stiff contact approximation.

2. Can FO-MBRL methods outperform MFRL in finding optimal policies? We introduce Adaptive Horizon Actor Critic (AHAC), a first-order model-based algorithm that mitigates gradient issues during stiff contact by adapting its trajectory rollout horizon (Figure 1). Experimentally, we show that it is capable of achieving up to 64% more asymptotic reward in comparison to model-free approaches across complex locomotion tasks.

3. Which methods are suitable for scaling to high-dimensional motor control tasks? We find that AHAC exhibits lower variance during training, offering stability and gradient accuracy, allowing it to scale effectively to high-dimensional motor control tasks with action dimension $\mathcal{A} = \mathbb{R}^{152}$.

## 2 Preliminaries

In this paper, we study discrete-time, finite-horizon, fully-observable reinforcement learning problems where the state of the system is defined as $\boldsymbol{s} \in \mathbb{R}^n$, actions are defined as $\boldsymbol{a} \in \mathbb{R}^m$, and the dynamics are governed by the function $f : \mathbb{R}^n \times \mathbb{R}^m \to \mathbb{R}^n$. Unlike traditional RL formulations, here we assume that the dynamics (i.e., transition function) are deterministic. At each timestep $t$, we sample an action from a stochastic policy

$\boldsymbol{a}_t \sim \pi_{\boldsymbol{\theta}}(\cdot|\boldsymbol{s}_t)$, which is parameterised by some parameters $\boldsymbol{\theta} \in \mathbb{R}^d$, and in turn, we receive a reward from $r : \mathbb{R}^n \times \mathbb{R}^m \to \mathbb{R}$. We can define the H-step return as:

$$R_H(\boldsymbol{s}_1, \boldsymbol{\theta}) = \sum_{h=1}^{H} r(\boldsymbol{s}_h, \boldsymbol{a}_h) \qquad s.t. \quad \boldsymbol{s}_{h+1} = f(\boldsymbol{s}_h, \boldsymbol{a}_h) \quad \boldsymbol{a}_h \sim \pi_{\boldsymbol{\theta}}(\cdot|\boldsymbol{s}_h)$$

As is typical in RL, the objective of the policy is to maximise the cumulative reward:

$$\max_{\boldsymbol{\theta}} J(\boldsymbol{\theta}) := \max_{\boldsymbol{\theta}} \mathbb{E}_{\substack{\boldsymbol{s}_1 \sim \rho \\ \boldsymbol{a}_h \sim \pi(\cdot|\boldsymbol{s}_h)}} [R_H(\boldsymbol{s}_1, \boldsymbol{\theta})] \tag{1}$$

where $\rho$ is the initial state distribution. Without loss of generality, we make our work easier to follow by making the following assumption:

**Assumption 2.1.** We assume that $\rho$ is a dirac-delta distribution.

Similar to prior work Duchi et al. (2012); Berahas et al. (2022); Suh et al. (2022), we are trying to exploit the smoothing properties of stochastic optimisation on the landscape of our optimisation objective. Following recent successful deep-learning approaches to MFRL (Schulman et al., 2017; Haarnoja et al., 2018), we assume:

**Assumption 2.2.** We assume that our policy is stochastic one parameterised by $\boldsymbol{\theta}$ and expressed as $\pi_{\boldsymbol{\theta}}(\cdot|\boldsymbol{s})$.

In order to solve our main optimisation problem in Equation 1, we consider using stochastic gradient estimates of $J(\boldsymbol{\theta})$. These can be obtained via zero-order and first-order methods. To guarantee the existence of $\nabla J(\boldsymbol{\theta})$, we need to make certain assumptions:

**Definition 2.3.** A function $g : \mathbb{R}^d \to \mathbb{R}^d$ has *polynomial growth* if there exists constants $a, b$ such that $\forall \mathbf{z} \in \mathbb{R}^d, ||g(\mathbf{z})|| \le a(1 + ||\mathbf{z}||^b)$.

**Assumption 2.4.** To ensure gradients are well defined, we assume that the policy $\pi_{\boldsymbol{\theta}}(\cdot|\boldsymbol{s})$ is continuously differentiable $\forall \boldsymbol{s} \in \mathbb{R}^n, \forall \boldsymbol{\theta} \in \mathbb{R}^d$. Furthermore, the system dynamics $f$ and reward $r$ have polynomial growth.

## 2.1 Zeroth-Order Batch Gradient (ZOBG) estimates

These weak assumptions are sufficient to make $J(\boldsymbol{\theta})$ differentiable in expectation by simply taking samples of the function value in a zeroth-order fashion. This gives us estimates of $\nabla J(\boldsymbol{\theta})$ via the stochasticity introduced by $\pi$, as first shown in (Williams, 1992), and commonly referred to as as the *Policy Gradient Theorem* (Sutton et al., 1999).

**Definition 2.5.** Given a sample of the H-step return $R_H(\boldsymbol{s}_1) = \sum_{h=1}^{H} r(\boldsymbol{s}_h, \boldsymbol{a}_h)$ following the policy $\pi$, we can estimate zero-order policy gradients via:

$$\nabla_{\boldsymbol{\theta}}^{[0]} J(\boldsymbol{\theta}) := \mathbb{E}_{\boldsymbol{a}_h \sim \pi_{\boldsymbol{\theta}}(\cdot|\boldsymbol{s}_h)} \left[ R_H(\boldsymbol{s}_1) \sum_{h=1}^{H} \nabla_{\boldsymbol{\theta}} \log \pi_{\boldsymbol{\theta}}(\boldsymbol{a}_h|\boldsymbol{s}_h) \right] \tag{2}$$

**Lemma 2.6.** Under Assumptions 2.1 and 2.4, the ZOBG is an unbiased estimator of the stochastic objective $\mathbb{E}\left[\bar{\nabla}^{[0]} J(\boldsymbol{\theta})\right] = \nabla J(\boldsymbol{\theta})$ where $\bar{\nabla}^{[0]} J(\boldsymbol{\theta})$ is the sample mean of $N$ Monte Carlo estimates of Eq. 2.

These zero-order policy gradients are known to have high variance, and one way to reduce their variance is by subtracting a baseline from the function estimates. Similar to (Suh et al., 2022), we do that by subtracting the return given by the noise-free policy rollout:

$$\nabla_{\boldsymbol{\theta}}^{[0]} J(\boldsymbol{\theta}) = \mathbb{E}_{\boldsymbol{a}_h \sim \pi_{\boldsymbol{\theta}}(\cdot|\boldsymbol{s}_h)} \left[ \left( R_H(\boldsymbol{s}_1) - R_H^*(\boldsymbol{s}_1) \right) \sum_{h=1}^{H} \nabla_{\boldsymbol{\theta}} \log \pi_{\boldsymbol{\theta}}(\boldsymbol{a}_h|\boldsymbol{s}_h) \right] \quad R_H^*(\boldsymbol{s}_1) := \sum_{h=1}^{H} r(\boldsymbol{s}_h, \mathbb{E}[\pi_{\boldsymbol{\theta}}(\cdot|\boldsymbol{s}_h)])$$

## 2.2 First-Order Batch Gradient (FOBG) estimates

Given access to a differentiable simulator (with contact approximation), one can directly compute the analytic gradients of $\nabla_{\boldsymbol{\theta}} R_H(\boldsymbol{s}_1)$ induced by the policy $\pi$:

$$\nabla_{\boldsymbol{\theta}}^{[1]} J(\boldsymbol{\theta}) := \mathbb{E}_{\boldsymbol{a}_h \sim \pi_{\boldsymbol{\theta}}(\cdot|\boldsymbol{s}_h)} [\nabla_{\boldsymbol{\theta}} R_H(\boldsymbol{s}_1)] \tag{3}$$

However, for these gradients to be well-defined, we need to make further assumptions:

**Assumption 2.7.** The system dynamics $f(\boldsymbol{s}, \boldsymbol{a})$ and the reward $r(\boldsymbol{s}, \boldsymbol{a})$ are continuously differentiable $\forall \boldsymbol{s} \in \mathbb{R}^n, \forall \boldsymbol{a} \in \mathbb{R}^m$.

## 3   Learning through contact

First-order gradients, as demonstrated in prior research, are asymptotically unbiased when $N \to \infty$ (Schulman et al., 2015). However, this ideal scenario is often impractical in real-world applications, leading to observed empirical bias, as indicated by (Suh et al., 2022). To illustrate this bias, we use the soft Heaviside function, a common tool for studying discontinuous functions in physics simulations as it is an approximation of the Coulomb friction model:

$$\bar{H}(x) = \begin{cases} 1 & x > \nu/2 \\ 2x/\nu & |x| \le \nu/2 \\ -1 & x < -\nu/2 \end{cases} \tag{4}$$

where $a \sim \pi_\theta(\cdot) = \theta + \mathcal{N}(0, \sigma^2)$. As shown in Appendix A, $\mathbb{E}_\pi\left[\bar{H}(a)\right]$ is a sum of error functions whose derivative $\nabla_\theta \mathbb{E}_\pi\left[\bar{H}(a)\right] \ne 0$ at $\theta = 0$. However, using FOBG, we obtain $\nabla_\theta \bar{H}(a) = 0$ in samples where $|a| > \nu/2$, which occurs with probability at least $\nu/\sigma\sqrt{2\pi}$. Since in practice we are limited in sample size, this translates to empirical bias that is inversely proportional to sample size, as shown in Figure 3. Notably, when $\nu \to 0$, we achieve a more accurate approximation of the underlying discontinuous function, but we also increase the likelihood of obtaining incorrect FOBG, thus amplifying bias in stochastic scenarios. We used this particular example to showcase empirical bias as our differentiable simulator used in Section 5 is based on the Coulomb friction model.

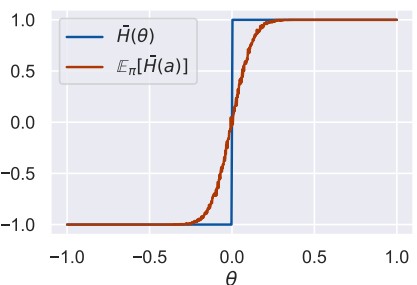

Figure 2: Soft Heavside of Eq 4.

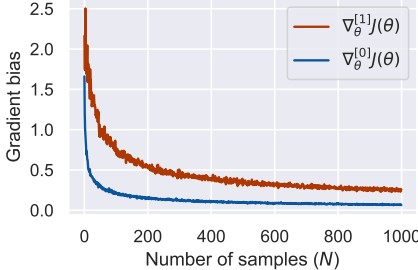

Figure 3: **Empirical bias of ZOBG and FOBG** when approximating the gradients of the Heaviside function.

By analysing the empirical bias from the perspective of bias and variance, we derive a practical upper bound:

**Lemma 3.1.** For an H-step stochastic optimisation problem under Assumptions 2.7, which also has Lipshitz-smooth policies $||\nabla\pi_{\boldsymbol{\theta}}(\boldsymbol{a}|\boldsymbol{s})|| \le B_\pi$ and Lipshitz-smooth and bounded rewards $r(\boldsymbol{s}, \boldsymbol{a}) \le ||\nabla r(\boldsymbol{s}, \boldsymbol{a})|| \le B_r$ $\forall \boldsymbol{s} \in \mathbb{R}^n; \boldsymbol{a} \in \mathbb{R}^m; \boldsymbol{\theta} \in \mathbb{R}^d$, then zero-order estimates remain unbiased. However, first-order gradient exhibit bias which is bounded by:

$$\left\| \mathbb{E}\left[\nabla_{\boldsymbol{\theta}}^{[1]} J(\boldsymbol{\theta})\right] - \mathbb{E}\left[\nabla_{\boldsymbol{\theta}}^{[0]} J(\boldsymbol{\theta})\right] \right\| \le H^4 B_r^2 B_\pi^2 \mathbb{E}_{\boldsymbol{a} \sim \pi}\left[\prod_{t=1}^{H} \|\nabla f(\boldsymbol{s}_t, \boldsymbol{a}_t)\|^2\right] \tag{5}$$

The proof can be found in Appendix B

We can ensure that the rewards are designed to meet the condition $r(\boldsymbol{s}, \boldsymbol{a}) \le ||\nabla r(\boldsymbol{s}, \boldsymbol{a})|| \le B_r$. The assumption over the policy $||\nabla\pi_{\boldsymbol{\theta}}(\cdot|\boldsymbol{s})|| \le B_\pi$ is less straightforward if we are using high-capacity models such as neural networks, but can be tackled via gradient normalisation techniques. However, giving any bounds over the dynamics $||\nabla f(\boldsymbol{s}_t, \boldsymbol{a}_t)||$ is difficult, yet influential in Equation 5. The Lemma also suggests that longer horizons results in exponential growth in bias.

To investigate the implications of Lemma 3.1, we constructed a simple scenario involving a ball rebounding off a wall and aiming to reach a target location, as illustrated in Figure 5. In this setup, the initial position $\boldsymbol{s}_1 = [x_1, y_1]$ and velocity of the ball are fixed. The objective is for the policy to learn the optimal initial orientation $\theta$ in order to reach a target position $\boldsymbol{s}_T$ at the end, defined as $R_H(\boldsymbol{s}_1) = \|\boldsymbol{s}_H - \boldsymbol{s}_T\|_2^{-1}$. Similar to before, we use an additive Gaussian policy $a = \theta + w$ where $w \sim \mathcal{N}(0, \sigma^2)$. Alternatively, $a \sim \pi_\theta(\cdot) = \mathcal{N}(\theta, \sigma^2)$

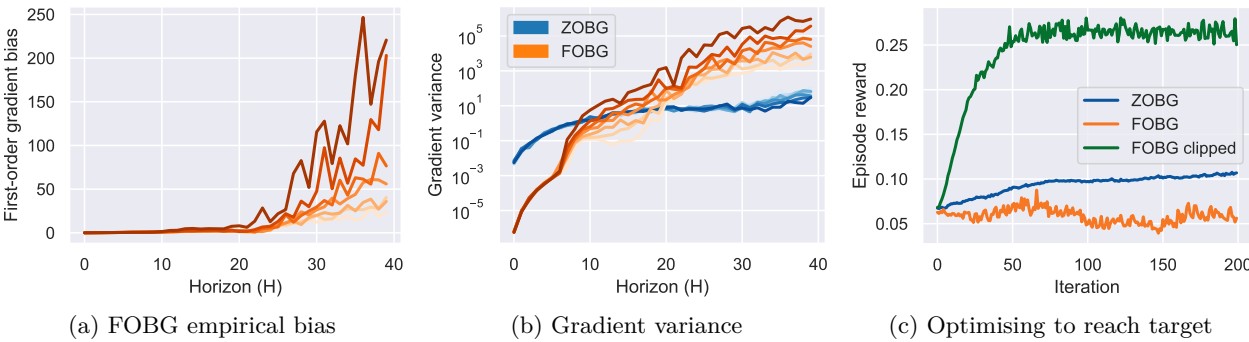

(a) FOBG empirical bias      (b) Gradient variance      (c) Optimising to reach target

Figure 4: **Results from the toy ball problem**. Fig (a) shows the empirical bias of FOBG measures by comparing it to ZOBG. Fig (b) shows how the variance of the two gradient types evolves over time. Both Figs (a) and (b) use different shades to show different stiffness configurations of the simulation with darker shades designating stiffer (and more realistic) simulation. Fig (c) shows attempts at optimising the initial angle of the ball to hit the target.

With this, zero-order gradients from Equation 2 can be expressed as:

$$\nabla_{\boldsymbol{\theta}}^{[0]} J(\theta) = \mathbb{E}_{\boldsymbol{a}_h \sim \pi(\theta)}\big[\big(R_H(\boldsymbol{s}_1) - R_H^*(\boldsymbol{s}_1)\big)\nabla \log \pi_\theta(a)\big] \approx \frac{1}{N\sigma^2}\big(R_H(\boldsymbol{s}_1) - R_H^*(\boldsymbol{s}_1)\big)w$$

We collect $N = 1024$ samples of each gradient type for each timestep with $H = 40$, starting from a randomly sampled starting angle $\theta \sim U(-\pi, \pi)$ for each environment. Figure 4a shows how the empirical bias of FOBG grows as $H$ increases, validating the proposed lemma. The bias remains low until the ball encounters contact, at which point it starts growing exponentially. We also examined the variance of the gradients in Figure 4b, observing that ZOBG follow $\mathrm{Var}\big[\nabla^{[0]} J(\theta)\big] \leq \frac{H B_r^2 B_\pi^2}{\sigma^2}$ first proposed by (Suh et al., 2022) - most importantly, they scale linearly with $H$. However, FOBG variance behaves similarly to the empirical bias bound described in Lemma 3.1.

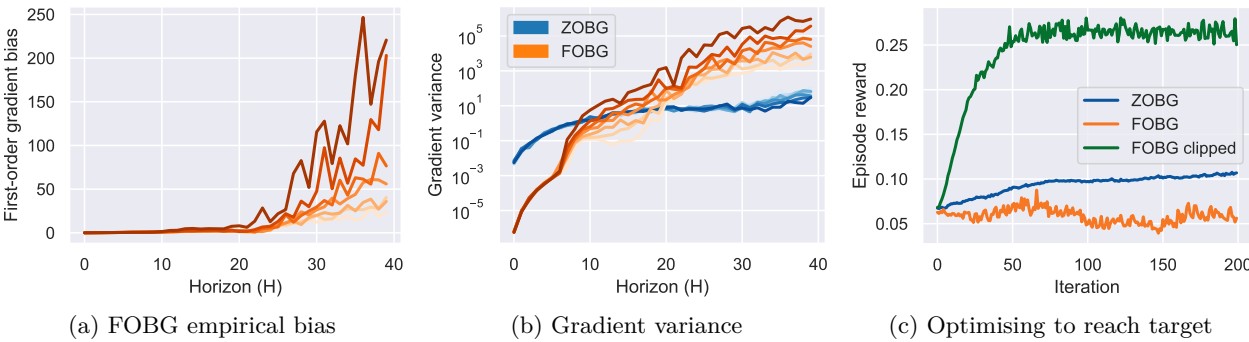

Figure 5: The toy problem where the ball is shot against a wall to reach the blue box.

exhibiting lower variance at the beginning of the rollout but growing exponentially. This is due mostly to the stiff dynamics $||\nabla f|| \gg 0$, which can be clearly seen at timestep $h = 6$ of Figure 4b where the ball first makes contact with the ground. Towards the end of the rollout, FOBG's variance can be up to five orders of magnitude higher than ZOBG, which remains unaffected by stiff dynamics. Both the bias and variance issues become even more pronounced as the contact stiffness increases, indicated by the darker shades in the figures.

This high empirical bias and the resulting high variance have significant consequences for optimisation and policy learning. We find that the biased FOBG fail to find a solution, as shown in Figure 4c. In contrast, the unbiased ZOBG have lower variance and slowly make progress towards a solution. This situation leads to a critical question: Is it possible to leverage the efficiency of FOBG in the presence of high bias gradients? Inspired by a common practice in deep learning, we attempt to normalise gradient norms of the dynamics. Employing this approach in Figure 4c shows that FOBG are now able to converge to a solution at a much faster rate than ZOBG.

$$\tilde{\nabla}_{\boldsymbol{s}_h} f(\boldsymbol{s}_h, \boldsymbol{a}_h) = gc(\nabla_{\boldsymbol{s}_h} f(\boldsymbol{s}_h, \boldsymbol{a}_h)) \quad \forall h \in [0, H]$$
$$\tilde{\nabla}_{\boldsymbol{a}_h} f(\boldsymbol{s}_h, \boldsymbol{a}_h) = gc(\nabla_{\boldsymbol{a}_h} f(\boldsymbol{s}_h, \boldsymbol{a}_h)) \quad \forall h \in [0, H]$$

$$gc(\boldsymbol{x}) := \begin{cases} \dfrac{\boldsymbol{x}}{||\boldsymbol{x}||_2} & \text{if}||\boldsymbol{x}||_2 > 1 \\ \boldsymbol{x} & \text{otherwise} \end{cases}$$

# 4 Adaptive Horizon Actor Critic (AHAC)

## 4.1 Learning through contact in a single environment

With a clearer understanding of stiff contact in differentiable simulators, we aim to develop a model-based algorithm employing FOBG for effective learning in infinite-horizon robotics tasks. Although we can apply the gradient clipping technique as above, in a multi-step problem, we can avoid the stiff gradients altogether. Consider a scenario where we have a 3-step trajectory ($H = 3$), and contact occurs at timestep $h = 3$, as shown in Figure 6. If we took the gradient normalisation approach, the gradients for $r(\boldsymbol{s}_3, \boldsymbol{a}_3)$ with a $\boldsymbol{\theta}$-parameterised policy where $\boldsymbol{a}_h \sim \pi_{\boldsymbol{\theta}}(\cdot|\boldsymbol{s}_h)$ with respect to $\boldsymbol{\theta}$:

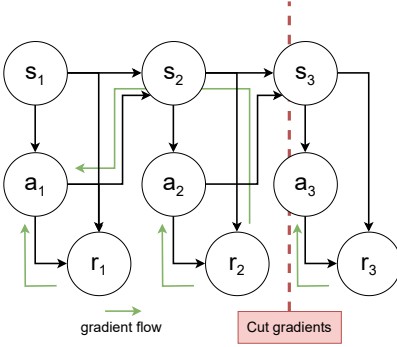

Figure 6: **H=3 trajectory with truncated gradients**. The green arrows indicate back-propagated gradients for *AHAC-1*.

$$
\begin{aligned}
\nabla_{\boldsymbol{\theta}} r(\boldsymbol{s}_3, \boldsymbol{a}_3) =& \nabla_{\boldsymbol{a}_3} r(\boldsymbol{s}_3, \boldsymbol{a}_3) \nabla_{\boldsymbol{\theta}} \pi_{\boldsymbol{\theta}}(\boldsymbol{a}_3|\boldsymbol{s}_3) \\
&+ \nabla_{\boldsymbol{s}_3} r(\boldsymbol{s}_3, \boldsymbol{a}_3) \tilde{\nabla}_{\boldsymbol{a}_2} f(\boldsymbol{s}_2, \boldsymbol{a}_2) \nabla_{\boldsymbol{\theta}} \pi_{\boldsymbol{\theta}}(\boldsymbol{a}_2|\boldsymbol{s}_2) \\
&+ \nabla_{\boldsymbol{s}_3} r(\boldsymbol{s}_3, \boldsymbol{a}_3) \tilde{\nabla}_{\boldsymbol{s}_2} f(\boldsymbol{s}_2, \boldsymbol{a}_2) \tilde{\nabla}_{\boldsymbol{a}_1} f(\boldsymbol{s}_1, \boldsymbol{a}_1) \nabla_{\boldsymbol{\theta}} \pi_{\boldsymbol{\theta}}(\boldsymbol{a}_1|\boldsymbol{s}_1)
\end{aligned}
$$

By definition of $gc(\boldsymbol{x})$ we will be losing gradient information. An alternative would be to cut gradients at $h = 2$. This would render $\nabla_{\boldsymbol{\theta}} r(\boldsymbol{s}_3, \boldsymbol{a}_3) = 0$, preventing us from learning in a scenario such as our toy example in Section 3. However, in a multi-step decision-making problem, $r(\boldsymbol{s}_3, \boldsymbol{a}_3)$ still yields gradients:

$$
\begin{aligned}
\nabla_{\boldsymbol{\theta}} \left[ \sum_{h=1}^{3} r(\boldsymbol{s}_h, \boldsymbol{a}_h) \right] =& \nabla_{\boldsymbol{a}_3} r(\boldsymbol{s}_3, \boldsymbol{a}_3) \nabla_{\boldsymbol{\theta}} \pi_{\boldsymbol{\theta}}(\boldsymbol{a}_3|\boldsymbol{s}_3) \\
&+ \nabla_{\boldsymbol{a}_2} r(\boldsymbol{s}_2, \boldsymbol{a}_2) \nabla_{\boldsymbol{\theta}} \pi_{\boldsymbol{\theta}}(\boldsymbol{a}_2|\boldsymbol{s}_2) + \nabla_{\boldsymbol{s}_2} r(\boldsymbol{s}_2, \boldsymbol{a}_2) \nabla_{\boldsymbol{a}_1} f(\boldsymbol{s}_1, \boldsymbol{a}_1) \nabla_{\boldsymbol{\theta}} \pi_{\boldsymbol{\theta}}(\boldsymbol{a}_1|\boldsymbol{s}_1) \\
&+ \nabla_{\boldsymbol{a}_1} r(\boldsymbol{s}_1, \boldsymbol{a}_1) \nabla_{\boldsymbol{\theta}} \pi_{\boldsymbol{\theta}}(\boldsymbol{a}_1|\boldsymbol{s}_1)
\end{aligned}
$$

Note how the gradients of dynamics in contact do not appear above. We call this technique *contact truncation*.

We present an FO-MBRL algorithm with an actor-critic architecture similar to SHAC (Xu et al., 2022). The critic, denoted as $V_{\psi}(\boldsymbol{s})$, is model-free and trained using TD($\lambda$) over an $H$-step horizon from timestep $t$:

$$
R_h(\boldsymbol{s}_t) := \sum_{n=t}^{t+h-1} \gamma^{n-t} r(\boldsymbol{s}_n, \boldsymbol{a}_n) + \gamma^{t+h} V_{\psi}(\boldsymbol{s}_{t+h}) \qquad \hat{V}(\boldsymbol{s}_t) := (1 - \lambda) \left[ \sum_{h=1}^{H-t-1} \lambda^{h-1} R_h(\boldsymbol{s}_t) \right] + \lambda^{H-t-1} R_H(\boldsymbol{s}_t)
$$

The critic loss becomes $\mathcal{L}_V(\boldsymbol{\psi})$, while the actor is trained using FOBG as in Equation 3, with the addition of the critic value estimate:

$$
\mathcal{L}_V(\boldsymbol{\psi}) := \sum_{h=t}^{t+H} \left\| V_{\psi}(\boldsymbol{s}_h) - \hat{V}(\boldsymbol{s}_h) \right\|_2^2 \qquad (6) \qquad J(\boldsymbol{\theta}) := \sum_{h=t}^{t+H-1} \gamma^{h-t} r(\boldsymbol{s}_h, \boldsymbol{a}_h) + \gamma^t V_{\psi}(\boldsymbol{s}_{t+T}) \quad (7)
$$

Unlike fixed-horizon model-based rollouts in (Xu et al., 2022), our policy is rolled out until stiff contact is encountered, which can be determined in simulation. This results in a dynamic FO-MBRL algorithm that adjusts its horizon to avoid exploding gradients. However, not all contact results in high bias; therefore, we want to truncate only on stiff contact $\|\nabla f(\boldsymbol{s}_t, \boldsymbol{a}_t)\| > C$, where $C$ is the contact stiffness parameter we set. We refer to this algorithm as Adaptive Horizon Actor Critic 1 (AHAC-1), which is designed for single environments but not suitable for vectorised environments (see Appendix D).

Using this approach, we can investigate if truncating gradients on contact yields better policy than naively cutting on fixed short-horizons. We compare SHAC and AHAC-1 on a simple contact-rich locomotion task, Hopper, a single-legged agent that obtains a reward for forward velocity (Figure 1). Both algorithms share the same hyperparameters, except for the ones related to horizons. SHAC uses a fixed $H = 32$, while AHAC-1 uses a maximum horizon of $H = 64$ with a contact threshold of $C = 500$. From Figure 7, we observe that AHAC-1 achieves a higher reward while exhibiting lower variance. Although difficult to analyse, we believe that our approach avoids local minima by adapting its horizon to avoid stiff gradients. On the other hand, SHAC gets pushed into local minima, which eventually results in policy collapse as seen in Figure 7. Unfortunately, AHAC-1 cannot be applied to parallel vectorised environments due to the challenge of asynchronously truncating trajectories, leading to infinitely long compute graphs.

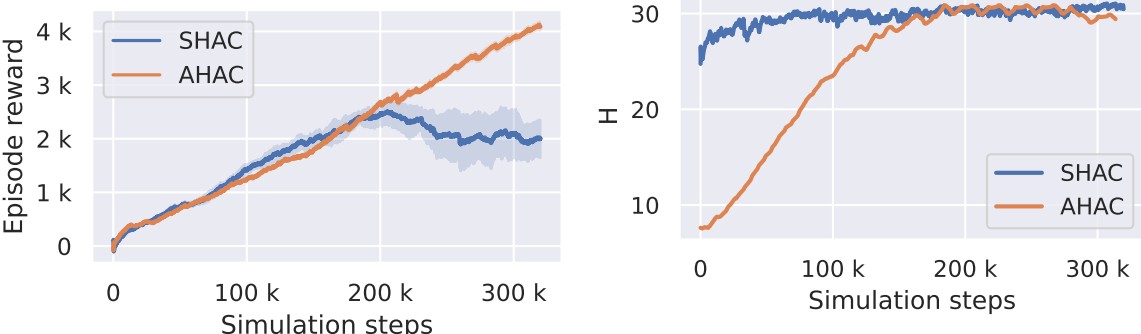

Figure 7: **Comparison between SHAC and AHAC-1 ran on the Hopper task with only a single environment**. The left plot shows rewards achieved by the algorithms over five different random seeds, with the mean and std. dev. plotted. The right plot is the moving window averaged mean horizon of both approaches. Note that even though SHAC has fixed horizons, they can still vary if the environment is terminated early due to early termination or episode end.

## 4.2 AHAC: A scalable approach for learning through contact

A straightforward solution to asynchronous truncation in *AHAC-1* might involve adopting the short-horizon approach of SHAC and truncating the graph on stiff contact. Unfortunately, this did not yield any performance improvements. Instead, we investigate the impact of the horizon length on policy optimality. We find that contact-based tasks have an inherent optimal solution, for instance, in a locomotion task, in the form of an optimal gait pattern. We observe that an SHAC agent converges to a particular solution, often suboptimal, depending on the horizon parameter $H$. Importantly, we find that asymptotic performance is maximised when the horizon $H$ matches the optimal (though unknown) gait frequency.

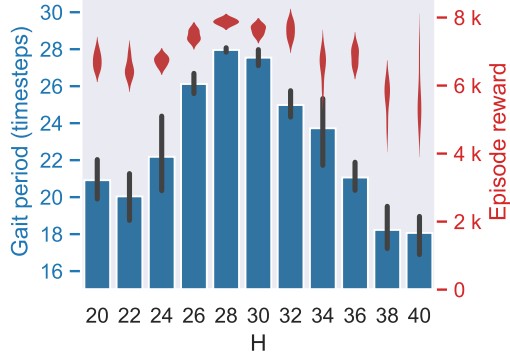

Figure 8: An ablation of short horizons $H$ for the SHAC algorithm applies to Ant. Each run is trained until convergence for 5 random seeds.

To empirically demonstrate this, we conducted an experiment by parameterizing SHAC with different $H$ values for Ant locomotion tasks, where a quadruped robot seeks to maximise forward velocity rewards. As seen from the results in Figure 8, the gait period aligns with the horizon length until $H = 28$, after which it attempts to fit two gait patterns within a single $H$-timestep rollout. Moreover, we noticed that the asymptotic reward reaches its peak as the horizon-length $H$ approaches what we believe to be the optimal gait period and displays the least variance across runs.

From these observations, we glean two insights: (1) each task has an optimal model-based horizon length $H$ that corresponds to the gait period, and (2) the associated optimal horizon results in the lowest variance between runs, supported by Lemma 3.1. We leverage these insights to generalise the AHAC-1 algorithm into a GPU-parallelisable approach, which we call AHAC. The critic training formulation remains the same as in Equation 6, but we introduce a new constrained objective for the actor:

$$J(\boldsymbol{\theta}) := \sum_{h=t}^{t+H-1} \gamma^{h-t} r(\boldsymbol{s}_h, \boldsymbol{a}_h) + \gamma^t V_{\boldsymbol{\psi}}(\boldsymbol{s}_{t+H}) \quad s.t. \quad \|\nabla f(\boldsymbol{s}_t, \boldsymbol{a}_t)\| \leq C \quad \forall t \in \{0, .., H\} \tag{8}$$

In simple terms, this objective seeks to maximise the reward while ensuring that all contact stiffness remains below a predefined threshold. Building on the inspiration from AHAC-1, we can progressively increase the

horizon as long as the constraint is satisfied. Using the Lagrangian formulation, we derive the dual problem:

$$\mathcal{L}_\pi(\boldsymbol{\theta}, \boldsymbol{\phi}) = \sum_{h=t}^{t+H-1} \gamma^{h-t} r(\boldsymbol{s}_h, \boldsymbol{a}_h) + \gamma^t V_{\boldsymbol{\psi}}(\boldsymbol{s}_{t+H}) + \boldsymbol{\phi}^T \left( \begin{bmatrix} \|\nabla f(\boldsymbol{s}_t, \boldsymbol{a}_t)\| \\ \vdots \\ \|\nabla f(\boldsymbol{s}_{t+H}, \boldsymbol{a}_{t+H})\| \end{bmatrix} - C \right) \tag{9}$$

By definition, $\phi_i = 0$ if the constraint is met and $\phi_i > 0$ otherwise. Thus, we can use $\boldsymbol{\phi}$ to adapt the horizon, resulting in the full AHAC algorithm shown in Algorithm 1. We train the critic until convergence is defined as a sufficiently small change in the last 5 critic training iterations, $\sum_{i=n-5}^{n} \mathcal{L}(\boldsymbol{\psi})$ where we take mini-batch samples from the buffer $(\boldsymbol{s}, \hat{V}(\boldsymbol{s})) \sim D$.

In practice, we find that truncating on $\nabla f(\boldsymbol{s}_t, \boldsymbol{a}_t)$ is limiting since different tasks involve varying contact forces, which often change throughout the learning process. Instead, we normalise the contact forces with modified acceleration per state dimension $\hat{\boldsymbol{q}}_t = \max(\boldsymbol{q}_t, 1)$ where the max is applied element-wise, resulting in normalised contact forces $\hat{\nabla} f(\boldsymbol{s}_t, \boldsymbol{a}_t) = \text{diag}(\hat{\boldsymbol{q}}_t) \nabla f(\boldsymbol{s}_t, \boldsymbol{a}_t)$. This allows us to use a single $C$ parameter across different tasks. Additionally, as contact approximation forces are computed separately in differentiable simulators, we don't need to utilise the full Jacobian of the dynamics. Instead, we can use the Jacobian derived only from contact. The differences between SHAC and AHAC are summarised in Appendix C.

---

**Algorithm 1:** Adaptive Horizon Actor-Critic

**Given**: $\gamma$: discount rate
**Given**: $\alpha$: learning rate
**Given**: $C$: contact threshold
**Initialise learnable parameters** $\boldsymbol{\theta}, \boldsymbol{\psi}, H, \boldsymbol{\phi} = \mathbf{0}$
$t \leftarrow 0$
**while** *episode not done* **do**
    /* rollout policy               */
    Initialise buffer  $D$
    **for**  $h = 0, 1, .., H$ **do**
      $\boldsymbol{a}_{t+h} \sim \pi_{\boldsymbol{\theta}}(\cdot|\boldsymbol{s}_{t+h})$
      $\boldsymbol{s}_{t+h+1} = f(\boldsymbol{s}_{t+h}, \boldsymbol{a}_{t+h})$
      $D \leftarrow D \cup \{(\boldsymbol{s}_{t+h}, \boldsymbol{a}_{t+h}, \boldsymbol{r}_{t+h}, V_{\boldsymbol{\psi}}(\boldsymbol{s}_{t+h+1}))\}$
    /* train actor with Eq. 9      */
    $\boldsymbol{\theta} \leftarrow \boldsymbol{\theta} + \alpha \nabla_{\boldsymbol{\theta}} \mathcal{L}_\pi(\boldsymbol{\theta}, \boldsymbol{\phi})$
    $\boldsymbol{\phi} \leftarrow \boldsymbol{\phi} + \alpha \nabla_{\boldsymbol{\phi}} \mathcal{L}_\pi(\boldsymbol{\theta}, \boldsymbol{\phi})$
    $H \leftarrow H - \alpha \sum_{t=0}^{H} \phi_t$
    /* train critic with Eq. 6      */
    **while** *not converged* **do**
      sample $(\boldsymbol{s}, \hat{V}(\boldsymbol{s})) \sim D$
      $\boldsymbol{\psi} \leftarrow \boldsymbol{\psi} - \alpha \nabla_{\boldsymbol{\psi}} \mathcal{L}_V(\boldsymbol{\psi})$
    $t \leftarrow t + H$

---

## 5 Experiments

In this section, we aim to address the following key questions experimentally:

1. Can first-order model-based (FO-MBRL) policies outperform zeroth-order model-free (ZO-MBRL) policies concerning asymptotoic episodic reward?

2. Does AHAC remain sample and wall-clock time efficient similar to prior FO-MBRL algorithms?

3. Does AHAC scale to high-dimensional environments?

4. Which components of AHAC contribute to its improved asymptotic reward compared to SHAC?

Previous FO-MBRL algorithms utilising differentiable simulators, such as SHAC (Xu et al., 2022), face challenges related to instability arising from stiff contact, which empirically results in worse asymptotic performance. The previous section and Figure 7 specifically suggest that the single-environment version of AHAC manages to avoid local minima and continue to progressively obtain a higher performance policy. We now investigate whether these benefits persist when scaling AHAC to $N = 512$ parallel environments and applying it to a more complex task: Ant, a quadruped with symmetrical legs, $\mathcal{S} = \mathbb{R}^{37}$ and $\mathcal{A} = \mathbb{R}^8$. The simulator used throughout this section is dflex, introduced by (Xu et al., 2022) and described in more detail in Appendix E. We compare AHAC to SHAC, its predecessor; PPO, a state-of-the-art on-policy MFRL algorithm (Schulman et al., 2017); SAC, an off-policy MFRL algorithm (Haarnoja et al., 2018); and SVG, a FO-MBRL method that does not utilise a differentiable simulator but instead learns its model of the dynamics[1]. A more explicit comparison of all of these approaches and more can be found in Section 6. We

---

[1]To make performance comparable, we attempted to vectorise SVG and found that it did not scale well with an increased number of parallel environments. Therefore, the results presented in this paper are from the original single-environment version.

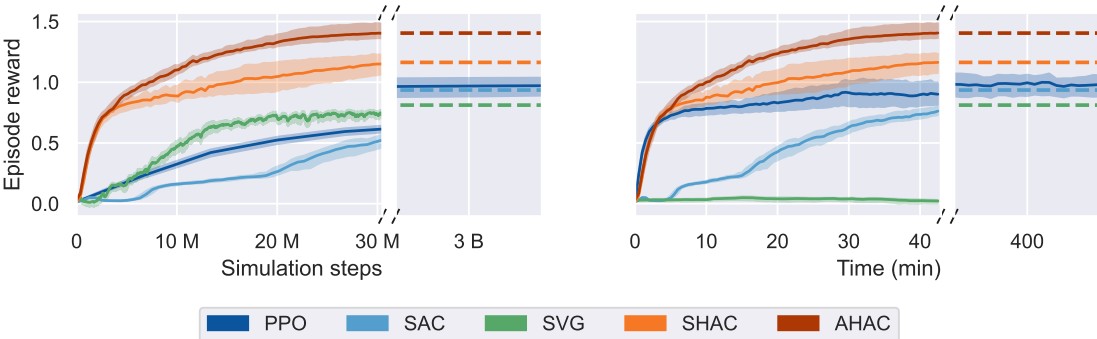

Figure 9: **Episodic rewards of the Ant task against both simulation steps and wall clock time**. The episodic reward is normalised by the highest mean reward achieved by PPO (i.e. PPO-normalised). The dashed lines represent the reward achieved by each respective algorithm at the end of their training runs.

tune and train all algorithms on the Ant task until convergence. Furthermore, we also train the MFRL baselines for 3B timesteps to investigate their asymptotic performance given practically infinite data. To best account for statistical errors given our limited runs, we employ the robust metrics suggested by (Agarwal et al., 2021). All results presented in this section show the 50% IQM and the 95% CI across 10 runs.

Figure 9 provides insights into the asymptotic performance. It shows that AHAC achieves a 41% higher reward than the closest model-free baseline, PPO, and outperforms SHAC with a smaller standard deviation between runs. Remarkably, the MFRL algorithms PPO and SAC get worse episodic rewards over time compared to AHAC, even when they are trained for 3B timesteps.

To answer questions (2) and (3) regarding efficiency and scalability, we conducted experiments across a variety of locomotion tasks. We again compare AHAC against SHAC, PPO, SAC, and SVG. We tune our model-free baselines sufficiently per-task to maximise performance. Due to long training times, SVG utilises similar hyper-parameters as in the original work (Amos et al., 2021). SHAC is tuned per-task but uses $H = 32$ across all tasks (Xu et al., 2022). AHAC uses the same shared hyper-parameters as SHAC and only has a tuned horizon learning rate of $\alpha_\psi$ per task. The full hyper-parameter details can be found in Appendix F. The following tasks share the same basic reward of maximising forward velocity with action penalties:

1. **Hopper**, a single-legged robot jumping only in one axis with $\mathcal{S} = \mathbb{R}^{11}$ and $\mathcal{A} = \mathbb{R}^3$.

2. **Anymal**, a sophisticated quadruped with $\mathcal{S} = \mathbb{R}^{49}$ and $\mathcal{A} = \mathbb{R}^{12}$ modelled after (Hutter et al., 2016).

3. **Humanoid**, a classic contact-rich environment with $\mathcal{S} = \mathbb{R}^{76}$ and $\mathcal{A} = \mathbb{R}^{21}$ which requires extensive exploration to find a good policy.

4. **SNU Humanoid**, a version of Humanoid lower body where instead of joint torque control, the robot is controlled via $\mathcal{A} = \mathbb{R}^{152}$ muscles intended to challenge the scaling capabilities of algorithms.

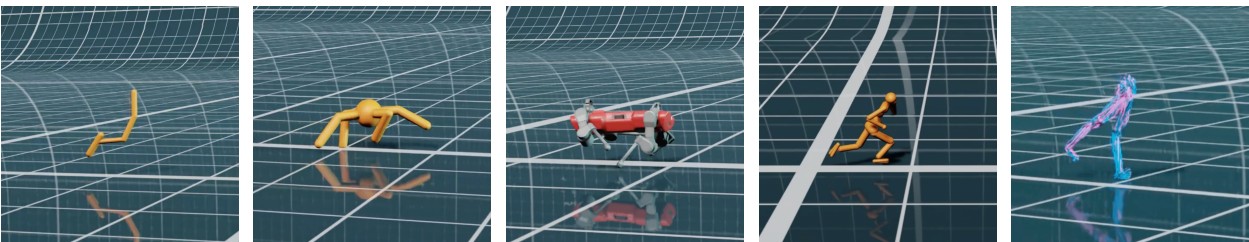

Figure 10: Locomotion environments (left to right): Hopper, Ant, Anymal, Humanoid and SNU Humanoid.

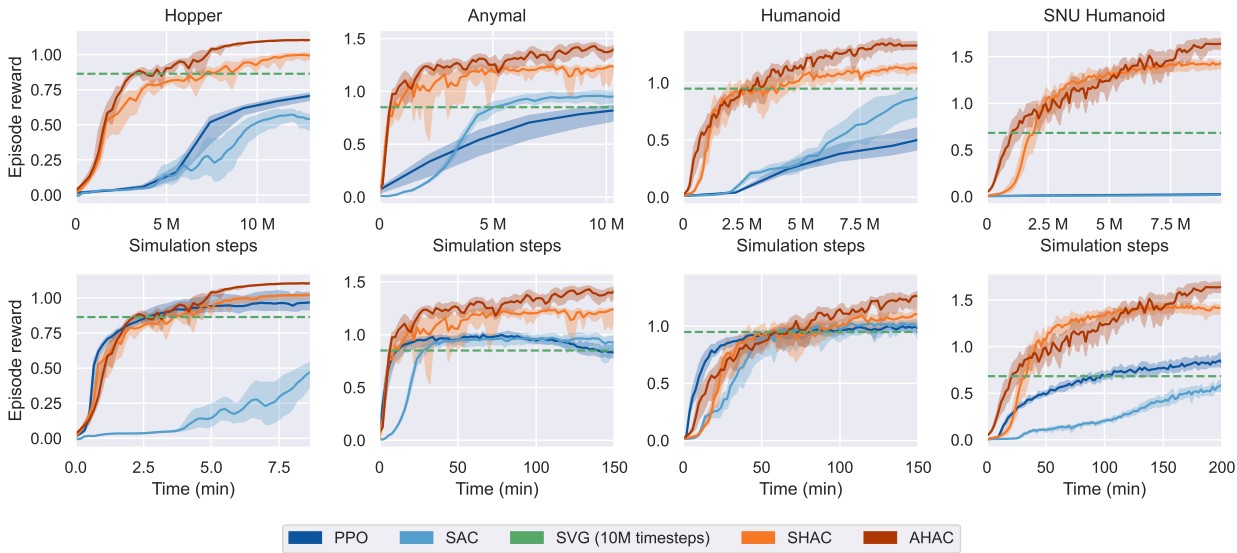

Figure 11: **Reward curves for all tasks against both simulation steps and training time**. The reward axis are normalised by the highest mean reward achieved by PPO. We further apply exponentially weighted smoothing with $\alpha = 0.98$ to increase legibility.

We compare the approaches against the number of simulation steps but also acknowledge that MFRL methods are computationally simpler and thus also provide results against wall-clock time. From the results in Figure 11, we observe that AHAC significantly outperforms the other methods while maintaining the sample efficiency of SHAC. On the simpler Hopper task, all algorithms achieve similar performance when compared against wall-clock time. However, with more complex tasks, we observe that AHAC not only learns at a faster rate but is also capable of achieving higher asymptotic performance across all tasks. This gap only becomes larger as we turn to more high-dimensional environments, showcasing the remarkable scalability of our approach, where AHAC achieves a 64% higher asymptotic reward than PPO on the SNU Humanoid task. Figure 12 shows aggregate summary statistics across all tasks, where AHAC outperforms all baselines but exhibits a larger confidence interval. Regardless, even at the tail end of the worst runs, AHAC achieves higher asymptotic rewards than our baselines, as seen in the score distributions. We also include tabular results in Appendix G.

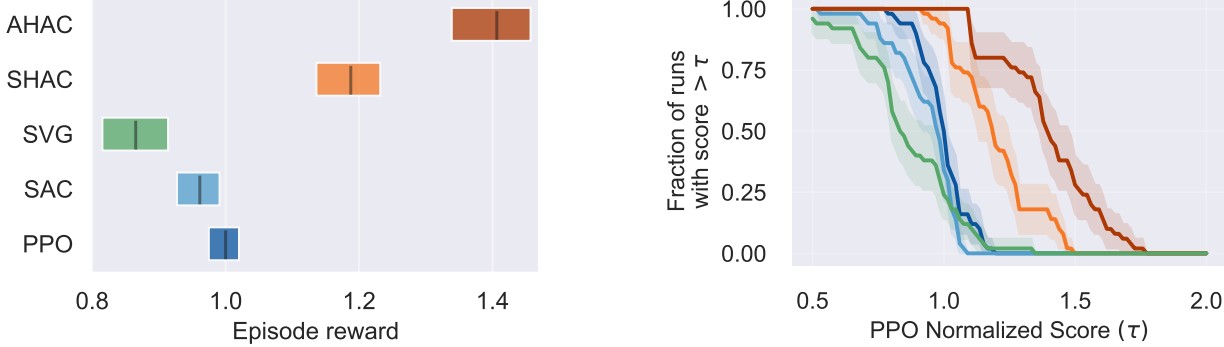

Figure 12: **Aggregate statistics comparing AHAC and our chosen baselines across all tasks.** The left figure shows 50% IQM of PPO-normalised reward with 95% CI. The right figure shows the score distributions of the algorithms as recommended in (Agarwal et al., 2021), which allows us to gauge the performance variability across tasks.

**Ablation study.** To understand the factors contributing to the performance demonstrated by AHAC, we investigate its key modifications, as detailed in Appendix C. Starting from our tuned baseline SHAC with $H = 32$, we add only one change per ablation:

1. SHAC H=29: using the $H$ converged by AHAC.
2. Adapt. Obj.: SHAC with Eq. 9 and fixed $H = 32$.
3. Adapt. Horizon: SHAC with Eq. 9 and adapting $H$.
4. Iterative critic: SHAC with iterative critic training.
5. Dual critic: SHAC with a dual critic.

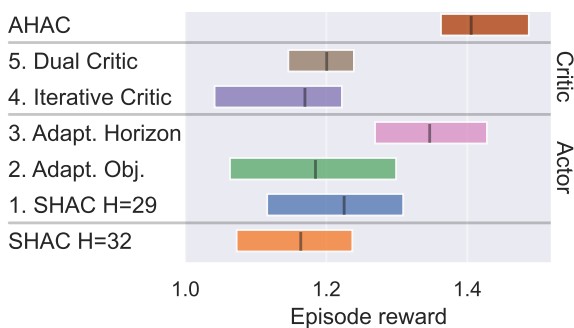

Figure 13: Ablations of AHAC on the Ant task.

All of the experiments use the same hyper-parameters tuned for SHAC, except for the horizon learning rate $\alpha_{\phi}$, which was tuned for AHAC. It is worth noting that SHAC with adaptive horizon (3) is equivalent to AHAC with single critic and no critic training until convergence. The outcomes presented in Figure 13 elucidate that the adaptive horizon objective notably enhances asymptotic reward. Rather surprisingly, SHAC with $H = 29$ achieves a higher reward than the baseline but fails to match the performance of the adaptive horizon mechanism. Similar to Section 4.1, we hypothesise that even when SHAC is using the converged optimal $H = 29$, it is still prone to getting stuck in local minima as the horizon might not be optimal during training. Simultaneously, the dual critic strategy substantially mitigates variance between runs compared to the target-critic employed in SHAC. Further ablation details and figures are provided in Appendix H.

## 6 Related work

In this section, we provide an overview of recent continuous control reinforcement learning (RL) methods, all of which follow the actor-critic paradigm (Konda & Tsitsiklis, 1999). The critic estimates the value of state-action pairs $Q(\boldsymbol{s}, \boldsymbol{a})$, and the actor learns optimal actions via $\max_{\boldsymbol{a}} Q(\boldsymbol{s}, \boldsymbol{a})$. While this section is intended as a review of related work, we also attempt to classify methods by their method of policy (actor) training, value estimator (critic) training, and their dynamics model $f(\boldsymbol{s}, \boldsymbol{a})$.

When no dynamics model is assumed, we are restricted to Model-Free Reinforcement Learning (MFRL) methods. We can take Monte-Carlo samples of the Policy Gradients Theorem to find $\nabla_{\boldsymbol{\theta}} J(\boldsymbol{\theta})$ using Equation 2. This allows MFRL methods to learn a feedback policy that predicts the distribution of actions given the state. On-policy methods, like Proximal Policy Optimisation (PPO) (Schulman et al., 2017), learn using only the most recent samples following the policy. In contrast, off-policy approaches, such as Soft Actor-Critic (SAC) (Haarnoja et al., 2018), can use all previously collected data at the expense of memory requirements.

Alternatively, Model-Based Reinforcement Learning (MBRL) methods aim to leverage a model for learning. This model can be learned from experience data or assumed a priori. In a basic scenario, it serves as an additional source of return estimates for the critic, which can still be trained in a model-free manner (Janner et al., 2019). Alternatively, the model can be used to obtain simulated returns for the critic, which can be first-order back-propagated through, known as Model-based Value Expansion (MVE) (Feinberg et al., 2018). Actor training is more intricate in this context. It can be done using Policy Gradients augmented by model-generated data (Janner et al., 2019) or as part of a gradient-free planning actor (Hafner et al., 2019b). This family of approaches is termed Zeroth-Order MBRL (ZO-MBRL). Alternatively, the returns of trajectories can be used to backpropagate through the model (Hafner et al., 2019a; Byravan et al., 2020), and we refer to these methods as First-Order MBRL (FO-MBRL). Key recent work is summarised in Table 1.

Recent interest in differentiable simulation has given rise to several works that employ FOBG to optimise objectives by back-propagating through the dynamics model (Hu et al., 2019b; Liang et al., 2019; Huang et al., 2021; Du et al., 2021). Although not explicitly addressing RL, these works follow the idea of rolling out a trajectory under a policy and iteratively optimising it until convergence. This approach can be reformulated as a FO-MBRL algorithm, referred to as Back-Propagation-Through-Time (BPTT).

When employed for typical long episodic RL tasks, BPTT performs poorly due to a noisy optimisation landscape and exploding gradients. (Xu et al., 2022) proposes Short Horizon Actor-Critic (SHAC) to address

| | Algorithm | Policy Learning | Value Learning | Dynamics Model |
|---|---|---|---|---|
| MFRL | PPO (Schulman et al., 2017) | Zeroth-order | Model-free | - |
| | SAC (Haarnoja et al., 2018) | Zeroth-order | Model-free | - |
| ZO-MBRL | MBPO (Janner et al., 2019) | Zeroth-order | Model-free | Ensemble NN |
| | PlaNet (Hafner et al., 2019b) | Gradient-free | - | Probabilistic NN |
| FO-MBRL | MVE (Feinberg et al., 2018) | Zeroth-order | Model-based | Deterministic NN |
| | STEVE (Buckman et al., 2018) | Zeroth-order | Model-based | Probabilistic NN |
| | Dreamer (Hafner et al., 2019a) | First-order | Model-based | Probabilistic NN |
| | IVG (Byravan et al., 2020) | First-order | Model-free | Deterministic NN |
| | SAC-SVG (Amos et al., 2021) | First-order | Model-free | Deterministic NN |
| | BPTT | First-order | - | Differentiable sim. |
| | SHAC (Xu et al., 2022) | First-order | Model-free | Differentiable sim. |
| | AHAC (this paper) | First-order | Model-free | Differentiable sim. |

Table 1: Comparison between recent influential RL algorithms for continuous control. We classify these approaches into the MFRL, ZO-MBRL and FO-MBRL categories predominantly by the way policy(actor) is learned. Zeroth-order (policy gradient) methods are harnessing the gradient estimates following Equation 2 without the need of taking dynamics model gradients, while First-order methods differentiate the whole trajectory following Equation 3. Model-based Value Learning refers to methods that fall under the Model-based Value Expansion (MVE) approach (Feinberg et al., 2018).

these issues by (1) introducing a model-free critic that acts as a smooth surrogate of the reward-maximisation objective and (2) by employing short rollouts to avoid high and unstable policy gradients. When run in a massively parallel fashion, SHAC stands out as one of the few MBRL approaches that achieves comparable asymptotic performance to MFRL methods while also demonstrating significantly better sample efficiency.

## 7   Conclusion

Our study aimed to compare the asymptotic performance of conventional Zeroth-order Model-Free RL (MFRL) methods with First-Order Model-Based (FO-MBRL) methods in differentiable simulators. We assessed the difference between both types of gradients and derived Lemma 3.1 showing that first-order batch gradient (FOBG) empirical bias is upper-bounded by the stiffness of the dynamics. Unfortunately, contact-rich tasks exhibit such properties, translating to FOBG with high bias and unstable learning.

We explored this issue in a toy problem and then introduced an algorithm designed to mitigate the accumulation of gradient bias stemming from stiff dynamics by truncating trajectories upon contact. When applied to high-dimensional locomotion tasks, our proposed approach, Adaptive Horizon Actor-Critic (AHAC), achieved up to a 64% increase in asymptotic episodic rewards compared to state-of-the-art MFRL methods. Surprisingly, we found that even with near-infinite data, MFRL methods cannot solve tasks with similar asymptotic reward to our proposed method. Additionally, AHAC retained the advantages commonly observed in FO-MBRL approaches, including exceptional sample efficiency and scalability to higher-dimensional tasks. Notably, our approach demonstrated the ability to learn complex locomotion policies for a quadruped robot in as little as 10 minutes on a single GPU, paving the way for RL scalability.

While AHAC outperforms MFRL methods in asymptotic rewards, it necessitates the development of differentiable simulators, requiring substantial engineering effort. Thus, we cannot help but admire the simple yet capable capable model-free algorithms such as PPO. Despite this, the performance of our proposed method renders it promising for robotic applications. However, it is essential to acknowledge the sim2real gap, which requires further exploration. Our vision for the next phase involves applying FO-MBRL approaches to real robots in a closed-loop manner where simulations aid policy learning but continually adapt to match the real environment. Furthermore, we believe that our proposed approach, AHAC, still has room for improvement, in particular by truncating each parallel environment independently, instead of learning a uniform horizon.

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

## A  Heavisde example

This appendix provides additional details on the Heaviside example used to obtain intuition regarding FOBG bias in Section 3.

$$\bar{H}(x) = \begin{cases} 1 & x > \nu/2 \\ 2x/\nu & |x| \le \nu/2 \\ -1 & x < -\nu/2 \end{cases}$$

Under stochastic input $x \sim \pi_\theta(\cdot) = \theta + w$ where $w \sim \mathcal{N}(0, \sigma)$, we can obtain the expected value:

$$\begin{aligned}
\mathbb{E}_w\left[\bar{H}(x)\right] &= \int_{-\infty}^{\infty} \bar{H}(x)\pi_\theta(x)dx \\
&= -\int_{-\infty}^{-\nu/2} \pi_\theta(x)dx + \int_{-\nu/2}^{\nu/2} \frac{2x}{\nu}\pi_\theta(x)dx + \int_{\nu/2}^{\infty} \pi_\theta(x)dx \\
&= -\frac{1}{2}\operatorname{erfc}\left(\frac{\nu + 2\theta}{2\sqrt{2}\nu}\right) + \frac{1}{2}\operatorname{erfc}\left(\frac{\nu - 2\theta}{2\sqrt{2}\nu}\right) + \frac{\theta}{\nu}\operatorname{erf}\left(\frac{\nu - 2\theta}{2\sqrt{2}\sigma}\right) + \frac{\theta}{\nu}\operatorname{erf}\left(\frac{\nu + 2\theta}{2\sqrt{2}\sigma}\right) \\
&\quad + \frac{\sigma\sqrt{2}}{\nu\sqrt{\pi}}\left(\exp -\frac{(\nu + 2\theta)^2}{8\sigma^2} - \exp -\frac{(\nu - 2\theta)^2}{8\sigma^2}\right)
\end{aligned}$$

From the expectation, we can obtain the gradient w.r.t. the parameter of interest:

$$\begin{aligned}
\nabla_\theta \mathbb{E}_w\left[\bar{H}(x)\right] &= \frac{1}{\sqrt{2\pi}\sigma}\exp\left(\frac{-(\nu + 2\theta)^2}{8\sigma^2}\right) + \frac{1}{\sqrt{2\pi}\sigma}\exp\left(\frac{-(\nu - 2\theta)^2}{8\sigma^2}\right) + \frac{1}{\nu}\operatorname{erf}\left(\frac{\nu - 2\theta}{2\sqrt{2}\sigma}\right) + \frac{1}{\nu}\operatorname{erf}\left(\frac{\nu + 2\theta}{2\sqrt{2}\sigma}\right) \\
&\quad - \frac{\sqrt{2}\theta}{\sqrt{\pi}\nu\sigma}\exp\left(\frac{-(\nu - 2\theta)^2}{8\sigma^2}\right) + \frac{\sqrt{2}\theta}{\sqrt{\pi}\nu\sigma}\exp\left(\frac{-(\nu + 2\theta)^2}{8\sigma^2}\right) \\
&\quad - \frac{1}{\sqrt{2}\sigma\nu}\exp\left(-(\nu - 2\theta)^{1/4\sigma^2}\right)(\nu - 2\theta)^{1/4\sigma^2 - 1} - \frac{1}{\sqrt{2}\sigma\nu}\exp\left(-(\nu + 2\theta)^{1/4\sigma^2}\right)(\nu + 2\theta)^{1/4\sigma^2 - 1}
\end{aligned}$$

As seen from the equation above, the true gradient $\nabla_\theta \mathbb{E}_w\left[\bar{H}(x)\right] \neq 0$ at $\theta = 0$. However, using FOBG, we obtain $\nabla_\theta \bar{H}(a) = 0$ in samples where $|a| > \nu/2$, which occurs with probability at least $\nu/\sigma\sqrt{2\pi}$. Even though both ZOBG and FOBG are theoretically unbiased as $N \to \infty$, both exhibit empirical bias as shown in Figure 14

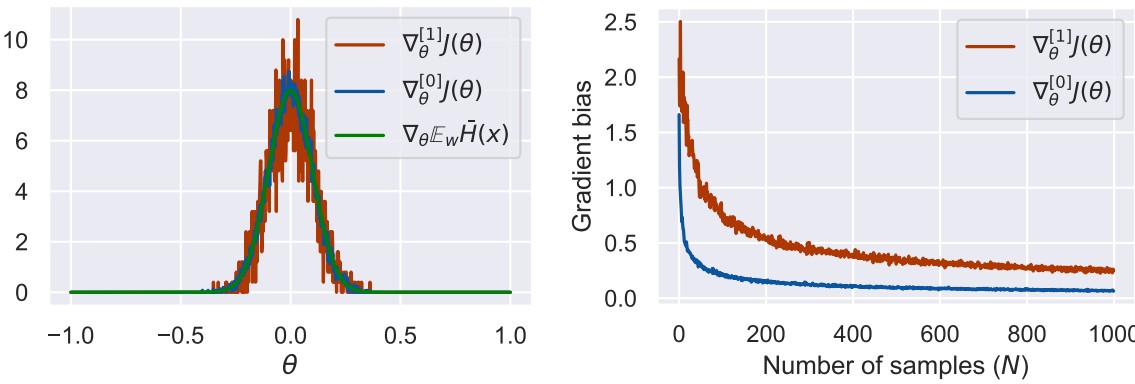

(a) Gradient estimates at $N = 1000$.  (b) Change in gradient bias for different sample sizes $N$.

Figure 14: Gradient bias study for the Soft Heaviside function shown in Eq. 4. Both ZOBG and FOBG exhibit bias at low samples sizes, however, FOBG are especially susceptible to the empirical bias phenomena.

# B  Proof of Lemma 3.1

**Assumption B.1.** As well as being continuously differentiable (Assumption 2.7, the policy is 1-Lipshitz-smooth: $\|\nabla_{\boldsymbol{\theta}} \pi_{\boldsymbol{\theta}}(\boldsymbol{a}|\boldsymbol{s})\| \leq B_{\pi} \leq 1$ and the reward function is 1 Lipshitz-smooth and bounded rewards $r(\boldsymbol{s}, \boldsymbol{a}) \leq \|\nabla r(\boldsymbol{s}, \boldsymbol{a})\| \leq B_r \leq 1 \ \forall \boldsymbol{s} \in \mathbb{R}^n; \boldsymbol{a} \in \mathbb{R}^m; \boldsymbol{\theta} \in \mathbb{R}^d.$

*Proof.* First, we expand our definition of bias and define a random variable of a single Monte-Carlo sample

$$\left\| \mathrm{Var}[\nabla]_{\boldsymbol{\theta}}^{[1]} J(\boldsymbol{\theta}) - \mathrm{Var}[\nabla]_{\boldsymbol{\theta}}^{[0]} J(\boldsymbol{\theta}) \right\| = \left\| \frac{1}{N} \sum_{i=1}^{N} \hat{\nabla}_{\boldsymbol{\theta}}^{[1]} J_i(\boldsymbol{\theta}) - \frac{1}{N} \sum_{i=1}^{N} \hat{\nabla}_{\boldsymbol{\theta}}^{[0]} J_i(\boldsymbol{\theta}) \right\|$$

$$= \frac{1}{N} \left\| \sum_{i=1}^{N} (\hat{\nabla}_{\boldsymbol{\theta}}^{[1]} J_i(\boldsymbol{\theta}) - \hat{\nabla}_{\boldsymbol{\theta}}^{[0]} J_i(\boldsymbol{\theta})) \right\| \tag{10}$$

Define $X_i = \hat{\nabla}_{\boldsymbol{\theta}}^{[1]} J_i(\boldsymbol{\theta}) - \hat{\nabla}_{\boldsymbol{\theta}}^{[0]} J_i(\boldsymbol{\theta})$ and bound it:

$$X_i = \sum_{h=1}^{H} \left( \nabla_{\boldsymbol{a}_h} r(\boldsymbol{s}_h, \boldsymbol{a}_h) \nabla_{\boldsymbol{\theta}} \pi_{\boldsymbol{\theta}}(\boldsymbol{a}_h|\boldsymbol{s}_h) + \sum_{h'=1}^{h-1} \nabla_{\boldsymbol{s}_h} r(\boldsymbol{s}_h, \boldsymbol{a}_h) \big( \prod_{t=1}^{h'} \nabla_{\boldsymbol{s}_t} f(\boldsymbol{s}_t, \boldsymbol{a}_t) \big) \nabla_{\boldsymbol{\theta}} \pi_{\boldsymbol{\theta}}(\boldsymbol{a}_{h'}|\boldsymbol{s}_{h'}) \right)$$

$$+ \sum_{h=1}^{H} r(\boldsymbol{s}_h, \boldsymbol{a}_h) \nabla \log \pi_{\boldsymbol{\theta}}(\boldsymbol{a}_h|\boldsymbol{s}_h)$$

$$= \sum_{h=1}^{H} \left( \nabla_{\boldsymbol{a}_h} r(\boldsymbol{s}_h, \boldsymbol{a}_h) \nabla_{\boldsymbol{\theta}} \pi_{\boldsymbol{\theta}}(\boldsymbol{a}_h|\boldsymbol{s}_h) - r(\boldsymbol{s}_h, \boldsymbol{a}_h) \nabla_{\boldsymbol{\theta}} \log \pi_{\boldsymbol{\theta}}(\boldsymbol{a}_h|\boldsymbol{s}_h) \right.$$

$$\left. + \sum_{h'=1}^{h-1} \nabla_{\boldsymbol{s}_h} r(\boldsymbol{s}_h, \boldsymbol{a}_h) \big( \prod_{t=1}^{h'} \nabla_{\boldsymbol{s}_t} f(\boldsymbol{s}_t, \boldsymbol{a}_t) \big) \nabla_{\boldsymbol{\theta}} \pi_{\boldsymbol{\theta}}(\boldsymbol{a}_{h'}|\boldsymbol{s}_{h'}) \right)$$

$$\leq \sum_{h=1}^{H} \sum_{h'=1}^{h-1} \nabla_{\boldsymbol{s}_h} r(\boldsymbol{s}_h, \boldsymbol{a}_h) \big( \prod_{t=1}^{h'} \nabla_{\boldsymbol{s}_t} f(\boldsymbol{s}_t, \boldsymbol{a}_t) \big) \nabla_{\boldsymbol{\theta}} \pi(\cdot, \boldsymbol{s}_{h'}) \qquad \text{(Assumption B.1)}$$

$$\leq \sum_{h=1}^{H} \sum_{h'=1}^{h-1} B_r B_{\pi} \prod_{t=1}^{h'} \|\nabla f(\boldsymbol{s}_t, \boldsymbol{a}_t)\|$$

We can now sum up these random variables as $Z = \sum_{i=1}^{N} X_i$ and create an upper concentration bound. As these RVs are difficult to bound, we can apply a Chebyshev Inequality (Tropp et al., 2015):

$$P(\|Z - \mathbb{E}[Z]\| > \epsilon) \leq \frac{\mathrm{Var}[Z]}{\epsilon^2}$$

Since the gradient samples are assumed to be i.i.d., we can expand this variance using the definition of each gradient type where all expectations are taken over the action distributions $\boldsymbol{a}_h$ for each step:

$$
\begin{aligned}
\mathrm{Var}[Z] = \mathrm{Var}\left[\sum_{i=1}^{N} X_i\right] &= \mathrm{Var}\left[\sum_{i=1}^{N} \hat{\nabla}_{\boldsymbol{\theta}}^{[1]} J_i(\boldsymbol{\theta}) - \hat{\nabla}_{\boldsymbol{\theta}}^{[0]} J_i(\boldsymbol{\theta})\right] \\
&= \sum_{i=1}^{N} \mathrm{Var}\left[\hat{\nabla}_{\boldsymbol{\theta}}^{[1]} J_i(\boldsymbol{\theta}) - \hat{\nabla}_{\boldsymbol{\theta}}^{[0]} J_i(\boldsymbol{\theta})\right] \\
&\leq \sum_{i=1}^{N} \mathbb{E}\left[\left\|\hat{\nabla}_{\boldsymbol{\theta}}^{[1]} J_i(\boldsymbol{\theta}) - \hat{\nabla}_{\boldsymbol{\theta}}^{[0]} J_i(\boldsymbol{\theta})\right\|^2\right] \\
&\leq \sum_{i=1}^{N} \mathbb{E}\left[\left\|\left\|\sum_{h=1}^{H}\sum_{h'=1}^{h-1} B_r B_\pi \prod_{t=1}^{h'} ||\nabla f(\boldsymbol{s}_t, \boldsymbol{a}_t)|| \;||^2\right\|\right] \\
&\leq N H^4 B_r^2 B_\pi^2 \, \mathbb{E}\left[\prod_{t=1}^{H} \|\nabla f(\boldsymbol{s}_t, \boldsymbol{a}_t)\|^2\right]
\end{aligned}
$$

With this result we can return back to Equation 10 and obtain

$$
\begin{aligned}
\left\|\mathrm{Var}[\nabla]_{\boldsymbol{\theta}}^{[1]} J(\boldsymbol{\theta}) - \mathrm{Var}[\nabla]_{\boldsymbol{\theta}}^{[0]} J(\boldsymbol{\theta})\right\| &\leq \frac{1}{N} N H^4 B_r^2 B_\pi^2 \, \mathbb{E}\left[\prod_{t=1}^{H} \|\nabla f(\boldsymbol{s}_t, \boldsymbol{a}_t)\|^2\right] \\
&= H^4 B_r^2 B_\pi^2 \, \mathbb{E}\left[\prod_{t=1}^{H} \|\nabla f(\boldsymbol{s}_t, \boldsymbol{a}_t)\|^2\right]
\end{aligned}
$$

$\square$

## C  Summary of modifications

To develop Adaptive Horizon Actor Critic (AHAC) algorithm, we used the Short Horizon Actor Critic (SHAC) algorithm (Xu et al., 2022) as a starting point. This section details all modifications applied to the SHAC in order to derive AHAC and achieve the reported results in this paper. We also note that some of these are not exclusive to either approach approach.

1. **Adaptive horizon objective** - instead of optimising the for the short horizon rollout return, we introduce the new constrained objective shown in Equation 8. To optimise that and adapt the horizon $H$, we introduced the dual problem in Equation 9 and optimised it directly for policy parameters $\boldsymbol{\theta}$ and the Lagrangian coefficients $\boldsymbol{\phi}$.

$$
J(\boldsymbol{\theta}) := \underbrace{\sum_{h=t}^{t+T-1} \gamma^{h-t} r(\boldsymbol{s}_h, \boldsymbol{a}_h) + \gamma^t V_{\boldsymbol{\psi}}(\boldsymbol{s}_{t+T})}_{\text{SHAC objective}} \quad s.t. \quad \|\nabla f(\boldsymbol{s}_t, \boldsymbol{a}_t)\| \leq C \quad \forall t \in \{0, .., T\}
$$
$$\underbrace{\phantom{XXXXXXXXXXXXXXXXXXXXXXXXXXXXXXXXXXXXXXXXXXXXXXXXXXXXXXXXXXXXXXXXXXXXXXXX}}_{\text{AHAC objective}}$$

2. **Dual critic** - the original implementation of SHAC struggled with more complex tasks such as Humanoid due the its highly non-convex value landscape. The authors of (Xu et al., 2022) solved that by introducing a delayed target critic similar to prior work in deep RL (Lillicrap et al., 2015). We found that approach brittle and requiring more hyper-parameter tuning. Instead, we replaced it with a dual critic similar to (Haarnoja et al., 2018) which has been shown to stabilise on-policy algorithms (Amos et al., 2021). For our work, we found that it reduced variance of asymptotic rewards achieved by AHAC while removing a hyperparameter.

3. **Critic training until convergence** - empirically we found that different problems present different value landscapes. The more complex the landscape, the more training the critic required and the critic often failed to fit the data with the limited number of critic iterations done in SHAC (16). Instead of training the critic for a fixed number of iterations, we trained the (dual) critic of AHAC until convergence defined by $\sum_{i=n-5}^{n} \mathcal{L}_i(\boldsymbol{\psi}) - \mathcal{L}_{i-1}(\boldsymbol{\psi}) < 0.5$ where $\mathcal{L}_i(\boldsymbol{\psi})$ is the critic loss for mini-batch iteration $i$. We allowed the critic to be trained for a maximum of 64 iterations. We found that this resulted in asymptotic performance improvements on more complex tasks such as Humanoid and SNU Humanoid while removing yet another hyper-parameter.

## D    AHAC-1 algorithm

---

**Algorithm 2:** Adaptive Horizon Actor-Critic (Single environment)

---

1  **Given**: $\gamma$: discount rate
2  **Given**: $\alpha$: learning rate
3  **Given**: $H$: maximum trajectory length
4  **Given**: $C$: contact threshold
5  **Initialise learnable parameters $\boldsymbol{\theta}, \boldsymbol{\psi}$**
6  $t \leftarrow 0$
7  **while** *episode not done* **do**
       /* rollout policy                                                                    */
8   |   Initialise buffer $D$
9   |   Initialise return $R \leftarrow 0$
10  |   **while** $\|\nabla f\| \leq C$ *and* $h \leq H$ **do**
11  |   |   $\boldsymbol{a}_t \sim \pi_{\boldsymbol{\theta}}(\cdot | \boldsymbol{s}_t)$
12  |   |   $s_{t+1} = f(\boldsymbol{s}_t, \boldsymbol{a}_t)$
13  |   |   $D \leftarrow D \cup \{(\boldsymbol{s}_{t+h}, \boldsymbol{a}_{t+h}, \boldsymbol{r}_{t+h}, V_{\boldsymbol{\psi}}(\boldsymbol{s}_{t+h+1}))\}$
14  |   |   $t \leftarrow t + 1$
       /* train actor with Eq.   7                                                          */
15  |   $\boldsymbol{\theta} \leftarrow \boldsymbol{\theta} - \alpha \nabla_{\boldsymbol{\theta}} J(\boldsymbol{\theta})$
       /* train critic with Eq.   6                                                         */
16  |   **while** *not converged* **do**
17  |   |   sample $(\boldsymbol{s}, \hat{V}(\boldsymbol{s})) \sim D$
18  |   |   $\boldsymbol{\psi} \leftarrow \boldsymbol{\psi} + \alpha \nabla_{\boldsymbol{\psi}} \mathcal{L}(\boldsymbol{\psi})$

---

## E    Simulation details

The experimental simulator, dflex (Xu et al., 2022), employed in Section 5, is a GPU-based differentiable simulator utilizing the Featherstone formulation for forward dynamics. It employs a spring-damper contact model with Coulomb friction.

The dynamics function $f$ is modeled by solving the forward dynamics equations:

$$M\ddot{q} = J^T \mathcal{F}(q, \dot{q}) + c(q, \dot{q}) + \tau(q, \dot{q}, a)$$

where, $q, \dot{q}, \ddot{q}$ are joint coordinates, velocities, and accelerations, respectively. $\mathcal{F}$ represents external forces, $c$ includes Coriolis forces, and $\tau$ denotes joint-space actuation. Mass matrix $M$ and Jacobian $J$ are computed concurrently using one thread per-environment. The composite rigid body algorithm (CRBA) is employed for articulation dynamics, enabling caching of the matrix factorization for reuse in the backward pass through parallel Cholesky decomposition.

After determining joint accelerations $\ddot{q}$, a semi-implicit Euler integration step updates the system state $s = (q, \dot{q})$. Torque-based control is employed for simple environments, where the policy outputs $\tau$ at each

time-step. For further details, see (Xu et al., 2022). It is noted that dflex is no longer actively developed and has been succeeded by warp (Macklin, 2022).

The rewards used across all experiments are designed to maximise the forward velocity $v_x$:

| Environment | Reward |
|---|---|
| Hopper | $v_x + R_{height} + R_{angle} - 0.1\|\boldsymbol{a}\|_2^2$ |
| Ant | $v_x + R_{height} + 0.1R_{angle} + R_{heading} - 0.01\|\boldsymbol{a}\|_2^2$ |
| Anymal | $v_x + R_{height} + 0.1R_{angle} + R_{heading} - 0.01\|\boldsymbol{a}\|_2^2$ |
| Humanoid | $v_x + R_{height} + 0.1R_{angle} + R_{heading} - 0.002\|\boldsymbol{a}\|_2^2$ |
| Humanoid STU | $v_x + R_{height} + 0.1R_{angle} + R_{heading} - 0.002\|\boldsymbol{a}\|_2^2$ |

Table 2: Table of hyper-parameters for all algorithms bench-marked in Section 5.

We additionally use auxiliary rewards $R_{height}$ to incentivise the agent to, $R_{angle}$ to keep the agents' normal vector point up, $R_{heading}$ to keep the agent's heading pointing towards the direction of running and a norm over the actions to incentivise energy-efficient policies. For most algorithms, none of these rewards apart from the last one are crucial to succeed in the task. However, all of them aid learning policies faster.

$$R_{height} = \begin{cases} h - h_{term} & if\, h \geq h_{term} \\ -200(h - h_{term})^2 & if\, h < h_{term} \end{cases}$$

$$R_{angle} = 1 - \left(\frac{\theta}{\theta_{term}}\right)^2$$

$R_{angle} = \|\boldsymbol{q}_{forward} - \boldsymbol{q}_{agent}\|_2^2$ is the difference between the heading of the agent $\boldsymbol{q}_{agent}$ and the forward vector $\boldsymbol{q}_{agent}$. $h$ is the height of the CoM of the agent and $\theta$ is the angle of its normal vector. $h_t erm$ and $\theta_t erm$ are parameters that we set for each environment depending on the robot morphology. Similar to other high-performance RL applications in simulation, we find it crucial to terminate episode early if the agent exceeds these termination parameters. However, it is worth noting that AHAC is still capable of solving all tasks described in the paper without these termination conditions, albeit slower.

## F  Hyper-parameters

This section details all hyper-parameters used in the main experiments of Section 5. PPO and SAC, as our MFRL baselines, have been tuned to perform well across all tasks, including task-specific hyper-parameters. SVG has not been specifically tuned for all benchmarks due to time limitations but instead uses the hyper-parameters presented in (Amos et al., 2021).[2] SHAC is tuned to perform well across all tasks using a fixed $H = 32$ as in the original work (Xu et al., 2022). AHAC shares all of its common hyper-parameters with SHAC and only has its horizon learning rate $\alpha_\phi$ tuned per-task. The contact threshold $C$ and iterative critic training criteria did not benefit from tuning. Note that the dual critic employed by AHAC, uses the same hyper-parameters used by the SHAC critic. Therefore, we have left AHAC under-tuned in comparison to SHAC in order to make to highlight the benefits of the adaptive horizon mechanism presented in this work.

Table 3 shows common hyper-parameters shared between all tasks. While table 4 shows hyper-parameters specific to each problem. Where possible we attempted to use the hyper-parameters suggested by the original works, however, we also attempted to share hyper-parameters between algorithms to ease comparison. If a specific hyper-parameter is not mentioned, then it is the one used in the original work behind the specific algorithm.

---

[2]Tuning SVG proved difficult as we were unable to vectorise the algorithm resulting in up to 2-week training times. This made it difficult to tune for our benchmarks

|  | AHAC | SHAC | PPO | SAC | SVG |
|---|---|---|---|---|---|
| Mini-epochs |  | 16 | 5 |  | 4 |
| Batch size | 8 | 8 | 8 | 32 | 1024 |
| $\lambda$ | 0.95 | 0.95 | 0.95 |  |  |
| $\gamma$ | 0.99 | 0.99 | 0.99 | 0.99 | 0.99 |
| H - horizon |  | 32 | 32 |  | 3 |
| C - contact thresh. | 500 |  |  |  |  |
| Grad norm | 1.0 | 1.0 | 1.0 |  |  |
| $\epsilon$ |  |  | 0.2 |  |  |
| Actor $log(\sigma)$ bounds |  |  |  | (-5,2) | (-5,2) |
| $\alpha$ - temperature |  |  |  | 0.2 | 0.1 |
| $\lambda_\alpha$ |  |  |  | $10^{-4}$ | $10^{-4}$ |
| $|D|$ - buffer size |  |  |  | $10^6$ | $10^6$ |
| Seed steps | 0 | 0 | 0 | $10^4$ | $10^4$ |

Table 3: Table of hyper-parameters for all algorithms benchmarked in Section 5. These are shared across all tasks.

|  | Hopper | Ant | Anymal | Humanoid | SNU Humanoid |
|---|---|---|---|---|---|
| Actor layers | (128, 64, 32) | (128, 64, 32) | (256, 128) | (256, 128) | (512, 256) |
| Actor $\alpha_\theta$ | $2 \times 10^{-3}$ | $2 \times 10^{-3}$ | $2 \times 10^{-3}$ | $2 \times 10^{-3}$ | $2 \times 10^{-3}$ |
| Horizon $\alpha_\phi$ | $2 \times 10^{-4}$ | $1 \times 10^{-5}$ | $1 \times 10^{-5}$ | $1 \times 10^{-5}$ | $1 \times 10^{-5}$ |
| Critic layers | (64, 64) | (64, 64) | (256, 128) | (256, 128) | (256, 256) |
| Critic $\alpha_\psi$ | $4 \times 10^{-3}$ | $2 \times 10^{-3}$ | $2 \times 10^{-3}$ | $5 \times 10^{e-4}$ | $5 \times 10^{-4}$ |
| Critic $\tau$ | 0.2 | 0.2 | 0.2 | 0.995 | 0.995 |

Table 4: Task-specific hyper-parameters. All benchmarked algorithms share the same actor and critic network hyper-parameters with ELU activation functions. AHAC and PPO do not have target critic networks and as such do not have $\tau$ as a hyper-parameter.

## G  Tabular experimental results

Below we present the asymptotic results of Section 5 in tabular form.

|  | Hopper | Ant | Anymal | Humanoid | SNU Humanoid |
|---|---|---|---|---|---|
| PPO | $1.00 \pm 0.11$ | $1.00 \pm 0.12$ | $1.00 \pm 0.03$ | $1.00 \pm 0.05$ | $1.00 \pm 0.09$ |
| SAC | $0.87 \pm 0.16$ | $0.95 \pm 0.08$ | $0.98 \pm 0.06$ | $1.04 \pm 0.04$ | $0.88 \pm 0.11$ |
| SVG | $0.84 \pm 0.08$ | $0.83 \pm 0.13$ | $0.84 \pm 0.19$ | $1.06 \pm 0.16$ | $0.75 \pm 0.23$ |
| SHAC | $1.02 \pm 0.03$ | $1.16 \pm 0.13$ | $1.26 \pm 0.04$ | $1.15 \pm 0.04$ | $1.44 \pm 0.08$ |
| AHAC | $1.10 \pm 0.00$ | $1.41 \pm 0.08$ | $1.46 \pm 0.06$ | $1.35 \pm 0.07$ | $1.64 \pm 0.07$ |

Table 5: Tabular results of the asymptotic (end of training) rewards achieved by each algorithm across all tasks. The results presented are 50 % IQM and standard deviation across 10 random seeds. All algorithms have been trained until convergence. The rewards presented are PPO-normalised.

## H  Ablation study details

In Section 5, we provided an ablation study of the individual contributions of our proposed approach, AHAC, as summarised in Appendix C. In this section, we provide further details on the conducted experiments. The aim of the study is to understand what changes contribute to the asymptotic performance of AHAC. To best

| | Ablation | H | Actor objective | Critic | Iterative critic training |
|---|---|---|---|---|---|
| | SHAC H=32 | 32 | Eq. 7 | Single w/ target | |
| Actor ablations | SHAC H=29 | 29 | Eq. 7 | Single w/ target | |
| | Adapt. Objective | 32 | Eq. 9 | Single w/ target | |
| | Adapt. Horizon | adaptive | Eq. 9 | Single w/ target | |
| Critic ablations | Iterative critic | 32 | Eq. 7 | Single w/ target | ✓ |
| | Dual critic | 32 | Eq. 7 | Dual | |
| | AHAC | adaptive | Eq. 9 | Dual | ✓ |

Table 6: Differences between ablations studied, split into actor and critic ablations. All ablations only introduce one component to the baseline, SHAC.

achieve that, we started from SHAC as the baseline using the tuned version detailed in Appendix F above. Afterwards we add the individual components that contribute to AHAC using the hyper-parameter from the section above. Note that only hyper-parameters particular to AHAC have been tuned to achieve the results presented in this paper; all other hyper-parameters are the ones tuned to our baseline SHAC with $H = 32$. In particular we have only tuned the adaptive horizon learning rate $\alpha_\psi$, contact threshold $C$ and the criteria for early stopping while doing iterative critic training. Table 6 shows the detailed differences between the ablations presented in Section 5. The ablations include:

1. SHAC H=32 - our baseline with most hyper-parameters tuned to it.

2. SHAC H=29 - SHAC using the horizon $H$ which AHAC converges to asymptotically.

3. Adapt. Objective - SHAC using the adaptive horizon objective introduced in Eq. 9 but without using it to adapt to horizon.

4. Adapt. Horizon - SHAC using the objective in Eq 9 and adapting the horizon. This is equivalent AHAC without the dual critic with iterative training.

5. Iterative critic - SHAC with a single target critic utilising iterative critic training until convergence.

6. Dual critic - SHAC with a dual critic and no target.

We provide ablations results on these changes on the Ant task in Figure 13. Wee also provide the learning curves for the same experiments in Figure 15 and tabular results in 7.

| Ablation | Asymptotic reward |
|---|---|
| SHAC H=32 | $1.16 \pm 0.14$ |
| SHAC H=29 | $1.23 \pm 0.17$ |
| Adaptive Objective | $1.18 \pm 0.18$ |
| Adaptive Horizon | $1.35 \pm 0.12$ |
| Iterative Critic | $1.17 \pm 0.13$ |
| Dual Critic | $1.20 \pm 0.07$ |
| AHAC | $1.41 \pm 0.08$ |

Table 7: Results of asymptotic performance of our ablation study showing 50% IQM and standard deviation.

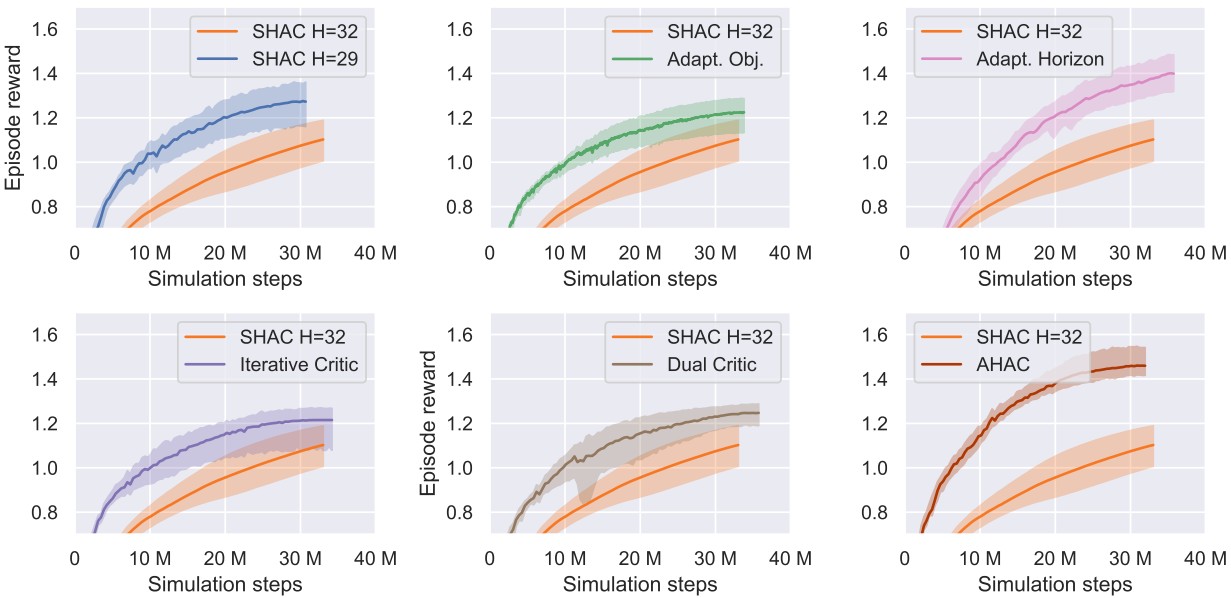

Figure 15: Standalone ablation results for the Ant task. These results are the same as in Figure 13 but presented in a different format for improved legibility.

