# OpenReview forum: "Learning optimal policies through contact in differentiable simulation"
_TMLR — Rejected by TMLR_

### Review · Reviewer_TQaX · 2023-12-02

**Summary Of Contributions:**

This paper proposes a way to exploit differentiable simulators in RL. It does so by truncating exploding gradients during contact, resulting in more stable gradient estimates and sample efficient learning. The algorithm is tested in locomotion tasks in MuJoCo, where it outperforms some other RL algorithms.

**Audience:**

Yes

**Claims And Evidence:**

Yes

**Requested Changes:**

I think the paper does a good job at following up intuitive explanation with thorough empirical evaluation. But, I think that the paper can be improved with some writing/presentation changes:
- The math in the paper was hard for me to follow, there were some abrupt jumps going from 1 step to the other.
- I was not familiar with the terminology of 1st order RL; it would be better to define that before talking about it in the Introduction.
- The SVG results were hard to make sense of, but I am sure you will update them when all your runs finish :)
- The biggest confusion I have is how the differentiable simulator is obtained and how you actually compute the gradient wrt state-action.

**Strengths And Weaknesses:**

Strengths:
- Overall approach is intuitive: truncate high-variance gradients that happen during contact.
- Evaluation is not limited to reward/learning curves, the paper specifically demonstrates high-mag gradients and that they can be mitigated.
- Learning curves show that the method is useful and promising for contact-rich tasks.
- I appreciated the effort behind the Related Work in Table 2.

Questions:
- How do you get access to a differentiable sim for the mujoco problems? Is that something you have to learn? I couldn't find that in the paper, apologies if I missed it.
- In Section 2, you assume that the transition function and start-state are deterministic. Are these assumptions necessary? I could not figure out how they are exploited in the main algorithm.
- In Section 3 Pg 4, you say that that the sq norm of the gradient of f is the biggest contributor to the variance in Eq 5; can you spell out how Eq 5 is a statement about variance?
- In the equation on top of Pg 5 (simplification of the "0th order" PG), what happened to the log in gradient? If $\pi(\theta)=\theta + w$, shouldn't the gradient wrt theta be $(1+w)$? Does $w$ depend on $\theta$?
- In Sec 4.1, you say that "SHAC gets pushed to local minima, which eventually results in policy collapse." Do you have evidence for this claim?
- Can you update the SVG results in Table 1 so that is a fair comparison?

---

> ### Author Response · Authors · 2023-12-17
> **Response**
>
> Thank you for the thorough review! We address specific concerns below:
>
> > In Section 3 Pg 4, you say that that the sq norm of the gradient of f is the biggest contributor to the variance in Eq 5; can you spell out how Eq 5 is a statement about variance?
>
> Thank you for highlighting this typo. We meant to refer to bias, which is defined in Eq 5. We have addressed this in the updated draft.
>
>
> > In the equation on top of Pg 5 (simplification of the "0th order" PG), what happened to the log in gradient? If $\pi(\theta) = \theta + w$, shouldn't the gradient wrt $\theta$ be $(1+w)$? Does $w$ depend on $\theta$?
>
> We forgot to include the log in the equation which has been addressed in the updated draft. Now $\nabla \log \pi_\theta(a) = \dfrac{a-\theta}{\sigma^2} = \dfrac{w}{\sigma^2}$ as $\pi_\theta(\cdot)$ is a normal distribution $\sim \mathcal{N}(\theta, \sigma^2)$
>
>
> > The math in the paper was hard for me to follow, there were some abrupt jumps going from 1 step to the other.
>
> Yes, we do agree with this and thought long how we can make our findings easy to follow by the general RL community. However, we do not have any clear ideas on what to improve further. We would welcome any particular suggestions you might have.
>
> > I was not familiar with the terminology of 1st order RL; it would be better to define that before talking about it in the introduction.
>
> Can you please clarify this question? If you are referring to what 1st order RL is, then we believe that our introduction does a good job at highlighting the difference between 0-order and 1-st order methods. We are open to specific suggestions on how we can make it more clear.
>
> > The SVG results were hard to make sense of, but I am sure you will update them when all your runs finish :)
>
> Yes, thank you for reminding us of this. We have now updated the results.
>
> > The biggest confusion I have is how the differentiable simulator is obtained and how you actually compute the gradient wrt state-action.
>
> Thank you for highlighting this. We addressed it in our general response and added an appendix explaining differentiable simulation in our draft. The most concise way we like thinking about differentiable simulation is when a simulation is built in such a way that the dynamics are fully differentiable. In more layman terms, think of $s_{t+1} = f(s_t, a_t)$ as being built with basic PyTorch operations that are fully differentiable. More particular details include making naturally discontinuous contact also differentiable and resolving contact in a finite timestep fashion (which Mujoco, for example, does not do). Please let us know if you think it is worth adding this type of explanation directly to our paper.

---

> > ### Comment · Reviewer_TQaX · 2024-01-02
> >
> > Thank you for addressing my concerns and for updating the manuscript; the details are easier for me to follow now.

---

### Review · Reviewer_DU72 · 2023-12-05

**Summary Of Contributions:**

First, the paper shows first-order model-based reinforcement learning (FO-MBRL) approaches suffer from high empirical bias and identifies "stiff contact" as the cause of this bias.

Then, the paper introduces Adaptive Horizon Actor Critic (AHAC) as a solution to this.  AHAC is a FO-MBRL method that optimizes for the horizon to avoid stiff contact.  This is accomplished by bounding the norm of the gradient of the dynamics function along the sequence, which is optimized using a Lagrange multiplier.

Finally, in five simulated control tasks AHAC achieves higher asymptotic reward than model-free methods and lower variance/sensitivity FO-MBRL methods.  The largest of the tasks has a 152-dimensional action space demonstrating to some extent that AHAC can scale to high dimensional control problems.

The authors have made code available on GitHub.

**Audience:**

Yes

**Broader Impact Concerns:**

None.

**Claims And Evidence:**

No

**Requested Changes:**

The caption of table 1 implies that H converges (at least experimentally).  One
would expect that short horizon actor/critic with that particular H should perform at least as well, if not better, than AHAC.  This means some combination of:
  (a) H in AHAC has not converged,
  (b) The SHAC experiments are undertuned,
  (c) There are some helpful learning dynamics of AHAC to speed convergence
It would be beneficial to investigate this.

**Strengths And Weaknesses:**

Strengths:

Section 3, though a tad on the long side, does a good job of demonstrating the bias in the first-order gradient estimates and its effect on learning.  In particular, lemma 3.1 theoretically justifies the norm constraint imposed by AHAC.

Weaknesses:

AHAC is not really adaptive in a strong sense.  The horizon does change while learning, but it is not a function of, e.g., the starting state.  It is not clear (either intuitively or from evidence in the paper) why it should perform better than SHAC with a parameter sweep over the horizon.

The need for the dynamics and reward functions to be deterministic, known ahead-of-time, and differentiable seems to be prohibitive limiting the approach to certain simulated environments only.

Some of the claims in the paper are not fully supported:
  - Re: FO-MBRL "This bias is primarily driven by the high magnitude dynamical gradients"--the particular example demonstrates this, but this does not imply this is the case in all domains.
  - Re: Lemma 3.1 "However, giving any bounds over the dynamics ||grad f||**2 is difficult, yet is the biggest contributor to the variance in Equation 5"--There is an H**4 term in equation 5 as well.  It is not at all obvious the norm term is always the biggest contributor.
  - "We believe our approach avoids local minima by adapting its horizon to stiff gradients"--what is the evidence that supports this?

The paper is not self-consistent and loose/under-specified in places:
  - In the preliminaries it is stated the paper discusses finite-horizon tasks, but upon introducing the algorithm it switches to infinite-horizon tasks with discounted reward.
  - H is defined as a natural number, but in the optimization it is a real.  Despite this, the real H is used to, e.g., index into the state sequence.
  - hat V depends on H, but this dependence is not explicitly spelled out.  It is not obvious in the algorithm what H to use as, e.g., values are stored in the replay buffer that depend on the H at the time they are stored.

---

> ### Author Response · Authors · 2023-12-17
> **Response**
>
> > AHAC is not really adaptive in a strong sense. The horizon does change while learning, but it is not a function of, e.g., the starting state. It is not clear (either intuitively or from evidence in the paper) why it should perform better than SHAC with a parameter sweep over the horizon.
>
> Thanks for your thoughtful assessment, and we appreciate your observations. The single-environment version of our approach, AHAC-1 (Section 4.1), is indeed adaptive and dependent on the starting state of each rollout. Unfortunately, we cannot extend this to a vectorized environment setup, as it would require us to asynchronously end each parallel rollout. We could not solve this issue and maintain GPU acceleration. Therefore, we decided to create AHAC that is not truly adaptive per-environment but mimics the adaptive mechanism in a batched setting. We see that this design choice, which is technically less adaptive than AHAC-1, is still better than the fixed rollout length of SHAC, as shown by our results in Section 5.
>
> > Re: FO-MBRL "This bias is primarily driven by the high magnitude dynamical gradients"--the particular example demonstrates this, but this does not imply this is the case in all domains.
> > Re: Lemma 3.1 "However, giving any bounds over the dynamics ||grad f||2 is difficult, yet is the biggest contributor to the variance in Equation 5"--There is an H4 term in equation 5 as well. It is not at all obvious the norm term is always the biggest contributor.
>
> It is important to clarify that we do not claim stiff dynamics to be universally responsible for high bias. In our study, our focus was directed towards learning in contact-rich tasks, where we firmly posit that stiff dynamics contribute significantly to first-order bias. To provide a numerical context, typically, when $H<100$, $|| \nabla r(s,a) ||$ can be made to satisfy $|| \nabla r(s,a) || \leq 1$ and $|| \nabla \pi(a|s) || \leq 1$ can be satisfied via gradient normalisation. However, in cases of stiff contact, our dynamics gradients tend to be in the range of $10^3 - 10^5$.
>
> Acknowledging your point, we also recognise the substantial role played by $H^4$ in bias, and we have incorporated a comment on its importance at the end of page 4.
>
> > In the preliminaries it is stated the paper discusses finite-horizon tasks, but upon introducing the algorithm, it switches to infinite-horizon tasks with discounted reward.
>
> Unfortunately, carrying out theoretical analysis in infinite time is more difficult. Similar to other works, we opted to do the analysis in finite time, derive intuition from it, and apply it to infinite-time problems. [Konda et al. (1999)](https://proceedings.neurips.cc/paper_files/paper/1999/file/6449f44a102fde848669bdd9eb6b76fa-Paper.pdf), [Byravan et al. (2019)](https://arxiv.org/pdf/1910.04142.pdf), [Haarnoja et al. (2018)](https://arxiv.org/pdf/1812.05905.pdf) and [Suh et al. (2022)](https://proceedings.mlr.press/v229/suh23a/suh23a.pdf).
>
> > H is defined as a natural number, but in the optimisation it is a real. Despite this, the real H is used to, e.g., index into the state sequence.
>
> Indeed, your observation is accurate. While it is essential to consider $H \in \mathbb{R}$ for optimisation purposes, in practical applications, we round it for natural number usage. We appreciate your diligence in pointing this out. In response to your feedback, we have included a more detailed explanation of this aspect in the revised draft.
>
> > hat V depends on H, but this dependence is not explicitly spelled out. It is not obvious in the algorithm what H to use as, e.g., values are stored in the replay buffer that depend on the H at the time they are stored.
>
> Your observation raises a valid point of potential confusion. In the main loop of Algorithm 1, it is crucial to note that the buffers are initialised with a length of $H$. Given that $H$ remains constant throughout the iteration, implementing the training loop becomes as straightforward as in other MBRL or PPO-like methods. It's worth mentioning that $\hat{V}(s)$ does indeed depend on $H$, and we believe this aspect is adequately addressed in our $TD(\lambda)$ formulation. Nevertheless, we remain open to any specific ideas you may have for enhancing the clarity of our approach.

---

> ### Author Response · Authors · 2023-12-17
> **Response**
>
> > The caption of table 1 implies that H converges (at least experimentally). One would expect that short horizon actor/critic with that particular H should perform at least as well, if not better, than AHAC. This means some combination of: (a) H in AHAC has not converged, (b) The SHAC experiments are undertuned, (c) There are some helpful learning dynamics of AHAC to speed convergence It would be beneficial to investigate this.
>
> The reviewer rightly points out that AHAC can be viewed as SHAC with an automatically tuned horizon. While it's accurate that SHAC with a tuned H hyperparameter may approximate AHAC's performance, tuning SHAC for complex problems is non-trivial and expensive. For example, running SHAC on the Ant task with horizons of H=38 and H=40 (Figure 8) results in similar performance with a variance equal to >50% of the mean. This stochasticity implies that the only reliable way to tune H is to run a hyper-parameter sweep, as we have done in Figure 8, which is expensive. Thus, we believe the value of AHAC and the automatically tuned horizon is immense.
>
> Addressing the additional question raised by the reviewer. H has indeed converged in all the AHAC experiments presented. While SHAC was tuned across all environments, it wasn't tuned on a per-environment basis, potentially making it under-tuned for specific environments. As noted in Appendix E, we use $H=32$ for SHAC.

---

> > ### Comment · Reviewer_DU72 · 2024-01-11
> >
> > > Unfortunately, we cannot extend this to a vectorized environment setup, as it would require us to asynchronously end each parallel rollout. We could not solve this issue and maintain GPU acceleration.
> >
> > It should be possible to vectorize it by unrolling the trajectories to match the longest length in the batch and masking out the gradients for the appropriate portions of the shorter trajectories.
> >
> > It still feels to me as if there is a disconnect between the paper's narrative and the AHAC algorithm.  In particular, if stiff dynamics are indeed a critical issue for FO-MBRL then the introduction of a state-agnostic H parameter seems artificially limiting.  This is perhaps more of a critique of SHAC, but it seems AHAC should not outperform SHAC with optimal hyperparameters.  i.e., at best AHAC saves one sweep over a marginally-beneficial parameter.
> >
> > > Addressing the additional question raised by the reviewer. H has indeed converged in all the AHAC experiments presented. While SHAC was tuned across all environments, it wasn't tuned on a per-environment basis, potentially making it under-tuned for specific environments.
> >
> > As mentioned by the action editor (though in a slightly different context), undertuning the baselines makes it harder interpret the experimental results.  It can also obscure important findings.  If the H parameter in AHAC indeed converges then it should never do better than SHAC in the limit, whereas the experiments imply it does better.  It's still possible that AHAC has favorable properties, e.g., perhaps it learns faster than SHAC with the best fixed H, but these many of these conclusions cannot be drawn from experiments without a strong baseline.
> >
> > If a hyperparameter sweep over H in SHAC is not feasible, perhaps a reasonable compromise may be to also show results when SHAC is given the H that AHAC converges to.

---

> > > ### Author Response · Authors · 2024-01-12
> > > **Response**
> > >
> > > Thank you for the insightful comments. We also had similar debates while doing this work and think that this is a good discussion to have.
> > >
> > > > It should be possible to vectorize it by unrolling the trajectories to match the longest length in the batch and masking out the gradients for the appropriate portions of the shorter trajectories.
> > >
> > > Certainly, this is possible, and we did attempt this using a maximum rollout horizon to avoid infinite trajectories (similar to AHAC-1). It is noteworthy that our empirical observations revealed a convergence towards asymptotic performance akin to AHAC. However, our investigation also brought to light suboptimal sample and wall-clock time efficiency.
> > >
> > > ![](https://i.imgur.com/ym7631C.png)
> > >
> > > We hypothesize this is due to our ineffective use of rollouts. A concrete illustration of this phenomenon can be delineated in a dual-environment scenario where the symbols `-` denote the backpropagation state, `o` signifies the contact state, and `x` represents the masked-out state, denoting instances where gradients are not leveraged. The trajectory rollout is exemplified as:
> > >
> > > ```
> > > --------o xxxxxxxx
> > > -----------------o
> > > ```
> > >
> > > illustrates a scenario where the gradients of the 'x' states are effectively neglected.
> > >
> > > Confronted with these empirical findings, we made a decision to develop AHAC as a pragmatic and scalable algorithm for adapting the horizon. It is essential to underscore that our conceptualization does not extend the foundational idea of AHAC-1 but rather seeks to emulate its principles in a practical manner.
> > >
> > > If you believe this is important to address in our paper, we would be happy to add an appendix detailing this issue further.
> > >
> > > > As mentioned by the action editor (though in a slightly different context), undertuning the baselines makes it harder interpret the experimental results. It can also obscure important findings. If the H parameter in AHAC indeed converges then it should never do better than SHAC in the limit, whereas the experiments imply it does better. It's still possible that AHAC has favorable properties, e.g., perhaps it learns faster than SHAC with the best fixed H, but these many of these conclusions cannot be drawn from experiments without a strong baseline.
> > >
> > > > If a hyperparameter sweep over H in SHAC is not feasible, perhaps a reasonable compromise may be to also show results when SHAC is given the H that AHAC converges to.
> > >
> > > In the most recent iteration of our manuscript, an ablation study has been incorporated, and Figure 12 presents asymptotic results for SHAC with $H=29$, as converged by AHAC. Although it is unclear why, it has come to our attention that AHAC consistently achieves superior results even in the absence of supplementary modifications to the critic, as indicated by the "Adapt. Horizon". Our underlying rationale posits that AHAC's efficacy lies in its ability to adeptly navigate away from local optima during training, thereby enabling a more reliable convergence toward a good policy.
> > >
> > > Nevertheless, it is imperative to underscore that, within the context of this specific ablation study, SHAC with $H=29$ does sporadically attain comparable results to AHAC, albeit infrequently (in this particular ablation, 1 out of 10 seeds).

---

### Review · Reviewer_fDX3 · 2023-12-07

**Summary Of Contributions:**

The manuscript introduces a new first order model-based reinforcement learning (MBRL) algorithm called Adaptive Horizon Actor-Critic (AHAC). This algorithm builds on top of a previous algorithm named Short-Horizon Actor-Critic (SHAC). The core idea behind these two algorithms is to carry out first order estimations of the gradient of the cumulative reward by means of differentiable dynamics and reward models, also termed as differentiable simulators.

Backpropagating through the dynamics and reward models appear to have a positive effect on the sample complexity, and even asymptotic performance, of these algorithms when compared with zeroth order model-free reinforcement learning (MFRL) algorithms.

The problem with naive backpropagation through dynamics and reward models is that first order approximations of the gradient are highly biased and present high variability. This bias is shown in the paper to be bounded by the dynamics models gradients. The dynamics gradients can be particularly large in states that are informally called contacts.

SHAC does this type of naive backpropagation and AHAC attempts to solve this by means of two different elements: a gradient clipping over the dynamics gradients, analogous to the gradient clipping observed in MFRL algorithms like PPO, and making adaptive a  hyperparameter that is fixed in SHAC and PPO called the horizon, which determines how large are the subtrajectories of an episode used to carry out backpropagation.

The motivation behind gradient clipping is that a smaller gradient will result in smaller bias and variance, which ideally would result in more stable deep reinforcement learning dynamics. Similarly, the idea behind having an adaptive horizon is that in this way backpropagation through transitions with high-normed gradients (contacts) would be eliminated altogether.

The way in which the horizon is adapted lies in the observation that, for continuous control environments with the main goal being locomotion, optimal policies display oscillatory behaviour, referred to as gait, with some natural period. This natural period coincides with the encounter of contacts and so with the desirable horizon to consider for backpropagation.

To instantiate the adaptive horizon, AHAC introduces a Lagrangian formulation of the cumulative reward maximization problem that requires gradients to have a small norm. Correspondingly, the horizon is updated by subtracting the dual variables associated to the dynamics gradients at each step. Intuitively, the dual variables will be 0 only for the natural period of gaits, and so, ideally, the adaptive horizon will correspond to this period after enough iterations.

Lastly, the experiments included in the manuscript suggest that AHAC outperforms alternative MBRL and MFRL methods in diverse locomotion tasks with high dimensional state and action spaces.

In summary, the manuscript proposes and motivates an MBRL algorithm that, based on the experiments, outperforms the sample efficiency and asymptotic cumulative reward of alternative MBRL and MFRL algorithms by improving naive backpropagation through contacts of a previous MBRL algorithm.

**Audience:**

Yes

**Broader Impact Concerns:**

I have no concerns.

**Claims And Evidence:**

No

**Requested Changes:**

Proposed adjustments:
1. Make modifications to the Introduction to make it more precise. What follows are some suggestions of points that could be detailed or modified (not critical):
- Make clear what is the distinction or relationship between simulators and models.
- Remove comments to the sim-to-real gap. This is not part of the motivation the paper develops later and raises questions about having perfectly deterministic dynamics.
- The use of first-order methods seems unmotivated when first mentioned.
- It is not clear what is the relationship between first-order methods and either "trajectory generation" in model-based control or "non-differentiable contact point discontinuities".
-What is the definition of brittle? Is it unrelated to "learning instability" or "hyperparameter tuning"? If not, it seems unnecessary to mention it.

2. Correct technical imprecisions or mistakes in Sections 2-5 (critical):
- $\hat{\nabla}^{[0]}J$ is not explicitly defined.
- Differentiable simulators are not defined.
- Contacts are not explicitly defined.
- In the Heaviside example, the policy should be a normal distribution with parameters, not with fixed mean 0 and deviation 1.
- In the same example, it is understandable but not clear what is the relationship between $R_1$, $\bar{H}$, $\theta$, $a$, $\sigma$, and $\nu$. Why not define $r(s,a)$ and $\theta$ explicitly?
- In the same example, the Heaviside function goes from 0 to 1, but in Figure 2 it goes from -1 to 1. It is not clear which one is incorrect since changing the 0 to -1 results in an expected value of 0, as opposed to what was stated. If -1 is changed to 0, then I do not get the same expected value either.
- In the same example, it is stated that "we achieve more realistic behaviour". What does this mean?
- In Figure 2, what is the orange line and the x-axis? The expected value is a scalar. How come it is represented by a curve?
- In Lemma 3.1, correct the first $s$ in $\|\|\nabla\pi_{\theta}(s|s)\|\|$.
- In Figure 4, the same color is being used for different things. In particular, FOBG clipped in c) has the same color as a high stiffness FOBG for a) and b), while there are no legends for a). I suggest changing the color of this curve in c) and possibly adding legends to a) to avoid confusions.
- I find it unnecessarily imprecise to state that the policy is a sum, right after Figure 4: $\pi(\theta)=\theta+w$. It is the action random variable the one that is equal to the sum of the mean and the noise, and the policy is a Gaussian with mean $\theta$ and deviation $\sigma$.
- In the Equation presented right after introducing the additive policy $\pi(\theta)=\theta+w$, the logarithm of the policy gradient disappeared, it is not clear where the approximation is coming from (or what it even means -- is this for large $N$?), and to me it makes no sense to have $w$ on the right hand side, since this is precisely the random variable with respect to to which the expected value would be taken.
- In this same ball example, it is not clear what is the relationship between $\psi$ and $a$. It is also not clear what it means to sample $\psi$ uniformly, given that this is, from what I understand, $a_1$.
- The reward in the ball example is introduced too late.
- At the end of Section 3 it is stated that "this preserves most of the gradient information". What does information mean in this setting? In order to say "most", the notion of information used should be measurable in some way. Is this the case?
- In subsection 4.1, I find the chain rule being employed confusing. It is correct, but not straightforward what it means to take the gradient of the policy provided that the action is not directly a function of the parameters. Why not explicitly define the random variable for action as a function of $\theta$?
- When defining the H-step TD($\lambda$) return, it probably should be $V_{\psi}$ instead of $V_{\phi}$.
- In Algorithm 1, since it is a maximization problem, the signs should be positive when updating $\theta$ and $\phi$.
- When defining the normalized "contact forces" (what is this?), what does it mean to divide 2 vectors that have different shapes?
- Stability (in question 1 of Section 5) is not defined.

3. Include ablation experiments (critical):
There are at least 6 characteristics of AHAC that make it different with respect to SHAC and the rest of the baseline models. The line of argumentation seems to suggest that 2 of them are central for the observed performance, but there is no evidence that the other 4 are not more important. Taking this into consideration, I consider it necessary to include ablation experiments that are conclusive regarding the relevance of the two main elements introduced by the paper. More specifically, the 6 elements I am referring to, 4 of which you already detail in the Appendix, are the following:
- adaptive horizon objective,
- clipped dynamics gradient norms,
- dual critic,
- critic training until convergence,
- dynamics gradient normalization ("normalised contact forces"),
- and TD($\lambda$) value estimation.

Then, I suggest making use of any experiment that shows that the performance difference between AHAC and SHAC lies mainly in elements 1 and 2. Regarding the 6th one, the comparison would be mainly with the other methods like PPO and SAC, where TD($\lambda$) is typically not used for the critic targets.

4. Add more seeds to all experiments (not critical)

**Strengths And Weaknesses:**

Strengths:
1. Overall, the paper does seem to introduce an interesting idea: first order backpropagation of the cumulative reward should consider an adaptive finite horizon that avoids large gradients. While not commented in the paper, this idea resembles the n-step TD and TD($\lambda$) estimates that can be used for the zeroth order methods.
2. The paper is very didactical in spirit, analyzing step-by-step the pitfall of SHAC and the possible solutions. In principle, this allows gaining an intuition of AHAC.
3. Although 5 seeds are a small number to make strong statistical conclusions, the experiments do seem to support positive answers to the questions 2-4 raised on Section 5 (for question 1, I see evidence for better sample efficiency, but not for "enhanced stability").

Weaknesses:
1. All being said, the imprecision (or straight errors) in terms and equations in the paper is so frequent that the motivations for AHAC end up being more confusing than enlightening.
2. Similarly, the use of so many techniques on top of SHAC makes unclear what exactly is making the algorithm perform so well. Specially since most of the techniques are heuristic in nature and tangential to the paper's line of argumentation.
3. A small number of seeds is being used per environment.

---

> ### Author Response · Authors · 2023-12-17
> **Response**
>
> Thank you for the thorough review! We appreciate the quality feedback and have acted on almost all of it. We address specific concerns below:
>
> > 1. Make modifications to the Introduction to make it more precise. What follows are some suggestions of points that could be detailed or modified (not critical):
> > Make clear what is the distinction or relationship between simulators and models.
> > Remove comments to the sim-to-real gap. This is not part of the motivation the paper develops later and raises questions about having perfectly deterministic dynamics.
> > The use of first-order methods seems unmotivated when first mentioned.
> > It is not clear what is the relationship between first-order methods and either "trajectory generation" in model-based control or "non-differentiable contact point discontinuities".
> > What is the definition of brittle? Is it unrelated to "learning instability" or "hyperparameter tuning"? If not, it seems unnecessary to mention it.
>
> Thank you for these comments. We have rewritten the Introduction in the updated draft. We think this contributed to making it more concise and easy to follow.
>
> > 2. Correct technical imprecisions or mistakes in Sections 2-5 (critical):
> > $\hat{\nabla}J$ is not explicitly defined.
>
> The hat notation simply designates an MC sample, as is common in statistical literature [(Suh et al. 2022)](https://arxiv.org/abs/2202.00817), [(Berahas et al. 2022)](https://arxiv.org/abs/1905.01332), [(Duchi et al. 2012)](https://arxiv.org/abs/1103.4296). We believe that is already clear in our text, but if further insistence is needed, we can clarify it.
>
> > Contacts are not explicitly defined.
>
> We are not sure what this question is referring to. As far as we are aware, there is no mathematical description of contact. If the reviewer is asking for a non-mathematical description, then contact is an event in which two bodies are in a state of interpenetration where we apply a contact model to resolve the physical phenomena.
>
> > In the same example, it is stated that "we achieve more realistic behaviour.". What does this mean?
>
> In this context, more realistic behaviour refers to a function approximation that is closer to the true underlying function. We added an explanation to this in our updated draft.
>
> > In Figure 2, what is the orange line and the x-axis? The expected value is a scalar. How come it is represented by a curve?
>
> It is the expected value wrt the added noise $w$ and is plotted against $\theta$ on the x axis. We have updated the notation in the figure to hopefully be more concise.
>
> > In Figure 4, the same color is being used for different things. In particular, FOBG clipped in c) has the same color as a high stiffness FOBG for a) and b), while there are no legends for a). I suggest changing the color of this curve in c) and possibly adding legends to a) to avoid confusions.
>
> Thank you for highlighting this. We have addressed 4c but we think that 4a is clear enough from the y-axis label and consistent colour coding.
>
> > I find it unnecessarily imprecise to state that the policy is a sum, right after Figure 4: $\pi(\theta) = \theta + w$. It is the action random variable the one that is equal to the sum of the mean and the noise, and the policy is a Gaussian with mean $\theta$ and $\sigma$ deviation.
> > In the Equation presented right after introducing the additive policy $\pi(\theta) = \theta + w$, the logarithm of the policy gradient disappeared, it is not clear where the approximation is coming from (or what it even means -- is this for large $N$ ?), and to me it makes no sense to have $w$ on the right hand side, since this is precisely the random variable with respect to to which the expected value would be taken.
>
> Thank you for highlighting the missing $\log$. We have added it but the right-hand side of the equation remains correct. We use the additive Gaussian notation precisely to make this simplification: $\nabla_\theta \pi_\theta(a) = \dfrac{a-\theta}{\sigma^2} = \dfrac{w}{\sigma^2}$.
>
> > In this same ball example, it is not clear what is the relationship between $\theta$ and $\psi$. It is also not clear what it means to sample $\psi$ uniformly, given that this is, from what I understand, .
> > The reward in the ball example is introduced too late.
>
>  Thank you for highlighting this. We have made modifications to address these concerns.
>
>
> > At the end of Section 3 it is stated that "this preserves most of the gradient information". What does information mean in this setting? In order to say "most", the notion of information used should be measurable in some way. Is this the case?
>
> We are not aware of a method to measure information in this case and we agree that our statement was imprecise. Thus we have decided to remove it.

---

> > ### Comment · Reviewer_fDX3 · 2024-01-05
> > **Response's response**
> >
> > Thank you for your thorough reply!
> >
> > I will answer to those replies I feel are still not satisfactory to me:
> >
> > > Thank you for these comments. We have rewritten the Introduction in the updated draft. We think this contributed to making it more concise and easy to follow.
> >
> > Thanks for the modifications to the Introduction. It is clearer and more engaging to me now. However, something that is still not clear to me is how using a simulator in addition to a MFRL method is not MBRL. As per the standard definition (see Sutton’s book), this is MBRL.
> >
> > > The hat notation simply designates an MC sample, as is common in statistical literature (Suh et al. 2022), (Berahas et al. 2022), (Duchi et al. 2012). We believe that is already clear in our text, but if further insistence is needed, we can clarify it.
> >
> > I disagree with the assertion that the hat notation is standard. It is usually used to denote statistical estimators, but I would not expect it to mean specifically a 1 sample Monte Carlo estimation. I do think this should be explicitly defined for clarity (actually, I would not  even mention that  approximation if the N sample estimator is already included).
> >
> > > We are not sure what this question is referring to. As far as we are aware, there is no mathematical description of contact. If the reviewer is asking for a non-mathematical description, then contact is an event in which two bodies are in a state of interpenetration where we apply a contact model to resolve the physical phenomena.
> >
> > If contact refers to the definition provided, I think it is overly narrow and somehow disconnected to the method being proposed, *yet it is in the title*. I thought that contact was a broader term that could be mathematically defined as displaying dynamics where the gradient was “too” large. Think for example of an electric field that permeates space, but which only has some effective range. Then, an electric particle getting in this range will display some dynamics that will effectively look like a collision, but there are no “two bodies in a state of interpenetration”. There is only one body that is always interacting with a field.
> >
> > > Thank you for highlighting the missing. We have added it but the right-hand side of the equation remains correct. We use the additive Gaussian notation precisely to make this simplification: $\nabla_\theta \pi_\theta(a) = \dfrac{a-\theta}{\sigma^2} = \dfrac{w}{\sigma^2}$
> >
> > I still think the right hand side is not correct. Nor $a$ nor $w$ can appear on that side because they are random variables and an expected value on the left hand side.
> >
> > > Thank you for highlighting this. We have made modifications to address these concerns.
> >
> > With the updated notation, it makes no sense to me to sample $\theta$. As far as I understand, $w$ (or $a$) is the one that should be sampled.

---

> > > ### Author Response · Authors · 2024-01-08
> > > **Response**
> > >
> > > Thank you again for the valuable feedback. We do appreciate it and strongly believe it will results in a better overall paper!
> > >
> > > > Thanks for the modifications to the Introduction. It is clearer and more engaging to me now. However, something that is still not clear to me is how using a simulator in addition to a MFRL method is not MBRL. As per the standard definition (see Sutton’s book), this is MBRL.
> > >
> > > Although the simulation can be viewed as a model, in an MFRL setting (such as PPO), it is only used for interactive data generation. On the other hand, MBRL approaches (such as AHAC), directly use the simulator for computation, not just sampling. Although this is a fine-detailed difference when benchmarking in simulation, the difference becomes much more obvious in real-world applications. MFRL approaches can simply use data from real-world interactions, whereas MBRL approaches must have access to a model of the real world. As such, we believe our terminology is correct.
> > >
> > > > I disagree with the assertion that the hat notation is standard. It is usually used to denote statistical estimators, but I would not expect it to mean specifically a 1 sample Monte Carlo estimation. I do think this should be explicitly defined for clarity (actually, I would not even mention that approximation if the N sample estimator is already included).
> > >
> > > Thank you for further clarifying this. Indeed, the notation used is confusing here. As per your suggestion, we have opted to completely remove the hat notation and just mention that the bar denotes the sample mean of MC estimates.
> > >
> > > > I still think the right hand side is not correct. Nor $a$ nor $w$ can appear on that side because they are random variables and an expected value on the left hand side.
> > >
> > > To reach that derivation we follow the one from [Chapter 20 of Underactuated Robotics](https://underactuated.mit.edu/rl_policy_search.html):
> > >
> > > $ a \sim \theta + w$ where $w \sim \mathcal{N}(0, \sigma)$. The resulting pdf becomes $\pi_\theta(a) = \dfrac{1}{\sqrt{2\pi} \sigma} \exp{ (-\dfrac{1}{2} \dfrac{(a-\theta)^2}{\sigma^2}})$
> > >
> > > $$ \begin{align}
> > > \nabla_\theta \log \pi_\theta(a) &=\nabla_\theta \Big[ \dfrac{1}{\sqrt{2\pi} \sigma} - \dfrac{1}{2} \dfrac{(a-\theta)^2}{\sigma^2} \Big] \\\\
> > > &= -\dfrac{1}{2\sigma^2} 2(a-\theta) (-1) \\\\
> > > &= \dfrac{a-\theta}{\sigma^2} \\\\
> > > &= \dfrac{w}{\sigma^2}
> > > \end{align}$$

---

> ### Author Response · Authors · 2023-12-17
> **Response**
>
> > In subsection 4.1, I find the chain rule being employed confusing. It is correct, but not straightforward what it means to take the gradient of the policy provided that the action is not directly a function of the parameters. Why not explicitly define the random variable for action as a function of $\theta$?
>
> We are unclear what the question is asking? $a \sim \pi_\theta(\cdot|s)$ as such, $a$ is a sample of a RV defined by $\pi$ whose pdf is a function of $\theta$. Maybe this alternative notation is clearer:  $a \sim \pi(\cdot|s; \theta)$
>
> > When defining the H-step TD($\lambda$) return, it probably should be $V_\psi$ instead of $V_\phi$.
> > In Algorithm 1, since it is a maximization problem, the signs should be positive when updating $\theta$ and $\phi$.
>
> Thank you for highlighting these typos. They have been addressed.
>
> > When defining the normalized "contact forces" (what is this?), what does it mean to divide 2 vectors that have different shapes?
>
> Thank you for highlighting these mathematical inaccuracies. We abused the overloaded notation $a$ to signify both action and accelerations. We have rewritten the section to be correct and clear.
>
> > Stability (in question 1 of Section 5) is not defined.
>
> We agree that stability is an imprecise quantity and have now opted to use variance instead.
>
> > Include ablation experiments (critical): There are at least 6 characteristics of AHAC that make it different with respect to SHAC and the rest of the baseline models. The line of argumentation seems to suggest that 2 of them are central for the observed performance, but there is no evidence that the other 4 are not more important. Taking this into consideration, I consider it necessary to include ablation experiments that are conclusive regarding the relevance of the two main elements introduced by the paper. More specifically, the 6 elements I am referring to, 4 of which you already detail in the Appendix, are the following:
> > adaptive horizon objective,
> > clipped dynamics gradient norms,
> > dual critic,
> > critic training until convergence,
> > dynamics gradient normalization ("normalised contact forces"),
> > and TD($\lambda$) value estimation.
>
>
> Thank you for providing such valuable feedback. We fully acknowledge the merit of including an ablation study in our paper and have made corresponding additions to the experiment section. Specifically, we have introduced an ablation analysis involving the adaptive horizon objective, dual critic, and training until convergence, focusing on the Ant task. In alignment with this, we made the decision to completely remove the clipped dynamics gradient norm from our proposed approach. This adjustment was made considering that the technique did not yield significant benefits. Experiments have been updated accordingly.
>
> Regarding the dynamics gradient normalization, we opted not to conduct an ablation as it primarily serves as a technical detail facilitating the adaptability of AHAC across diverse tasks. In the context of a single task, it proves inconsequential.
>
> Lastly, as the SHAC paper [(Xu et al. (2022)](https://short-horizon-actor-critic.github.io/)) already includes an ablation study of TD($\lambda$) targets, we did not include a similar study in our work.
>
> We genuinely appreciate your feedback and believe that it has significantly enhanced the quality of our work.

---

> > ### Comment · Reviewer_fDX3 · 2024-01-05
> > **Response's response**
> >
> > > We are unclear what the question is asking? $a \sim \pi_\theta(\cdot|s)$ as such, $a$ is a sample of a RV defined by $\pi$ whose pdf is a function of $\theta$. Maybe this alternative notation is clearer: $a \sim \pi(\cdot|s; \theta)$.
> >
> > To me, it is technically imprecise to take the gradient of a random variable with respect to a parameter of the distribution associated to the random variable. However, I admit it is an unnecessary technicality. My fear is that this is not obvious for some readers.
> >
> > > Thank you for highlighting these mathematical inaccuracies. We abused the overloaded notation
> >  to signify both action and accelerations. We have rewritten the section to be correct and clear.
> >
> > I'm still not sure I understand the dimensionality here. Does the normalization only apply when the gradient is being taken with respect to the state?
> >
> > > Thank you for providing such valuable feedback. We fully acknowledge the merit of including an ablation study in our paper and have made corresponding additions to the experiment section. Specifically, we have introduced an ablation analysis involving the adaptive horizon objective, dual critic, and training until convergence, focusing on the Ant task. In alignment with this, we made the decision to completely remove the clipped dynamics gradient norm from our proposed approach. This adjustment was made considering that the technique did not yield significant benefits. Experiments have been updated accordingly.
> >
> > Based on my intuition of AHAC-1 and the comments made by reviewer DU72, I find the current ablation still insufficient to disentangle 2 different factors: the adaptability of the horizon itself and the penalization of the dynamics' gradients. The reason is that what you call "adaptive horizon" includes 2 different things: 1) $H$ changes according to $\phi$ and 2) $\nabla f$ appears in the Lagrangian $\mathcal{L}$. Thus, the ablation results indicate that the gain of AHAC over SHAC can be a consequence of any of these two factors.
> >
> > I suggest two different experiments: 1) fixing $H$ to the same value as for SHAC and seeing if the gain still holds for this "adaptive horizon" ablation, and 2) running SHAC with the final horizon found by AHAC. If 1) is positive and 2) is not (meaning SHAC still does worst), that means that, as reviewer DU72 partially suggests, SHAC should not have the same performance as AHAC in the limit since the learning dynamics are just different thanks to the penalization of large dynamics' gradients. Also, if this turns out to be the case, I would suggest reconsidering the name "AHAC".
> >
> > > Lastly, as the SHAC paper (Xu et al. (2022)) already includes an ablation study of TD($\lambda$) targets, we did not include a similar study in our work.
> >
> > I did not find this ablation in the reference you mention, but I do not find this case as relevant as the previous one regarding the "adaptive horizon".

---

> > > ### Author Response · Authors · 2024-01-08
> > > **Response**
> > >
> > > > I'm still not sure I understand the dimensionality here. Does the normalization only apply when the gradient is being taken with respect to the state?
> > >
> > > Normalization is applied to both gradients. Since $f: R^m \times R^n \rightarrow R^m$, the Jacobians wrt to state and action have dimensionalities $n \times n$ and $n \times m$ respectively. That enables us to apply normalization uniformly across both Jacobians.
> > >
> > > > Based on my intuition of AHAC-1 and the comments made by reviewer DU72, I find the current ablation still insufficient to disentangle 2 different factors: the adaptability of the horizon itself and the penalization of the dynamics' gradients. The reason is that what you call "adaptive horizon" includes 2 different things: 1) $H$ changes according to $\phi$ and 2) $\nabla f$ appears in the Lagrangian $\mathcal{L}$. Thus, the ablation results indicate that the gain of AHAC over SHAC can be a consequence of any of these two factors.
> > >
> > > > I suggest two different experiments: 1) fixing $H$ to the same value as for SHAC and seeing if the gain still holds for this "adaptive horizon" ablation, and 2) running SHAC with the final horizon found by AHAC. If 1) is positive and 2) is not (meaning SHAC still does worst), that means that, as reviewer DU72 partially suggests, SHAC should not have the same performance as AHAC in the limit since the learning dynamics are just different thanks to the penalization of large dynamics' gradients. Also, if this turns out to be the case, I would suggest reconsidering the name "AHAC".
> > >
> > > Thank you for this great suggestion. We have run ablation experiments and updated the ablation study at the end of page 10. It now includes (1) standard SHAC with $H=32$, (2) SHAC with the optimal horizon $H=29$ converged to by AHAC (and supported by results in Fig. 8), (3) SHAC + AHAC objective but without learning $H$ and (4) SHAC + AHAC objective and learning $H$. As indicated by the results, there is still substantial value to AHAC and its approach to learning the horizon.
> > >
> > > > I did not find this ablation in the reference you mention, but I do not find this case as relevant as the previous one regarding the "adaptive horizon".
> > >
> > > Figure 4 of [(Xu et. al. 2022)](https://arxiv.org/abs/2204.07137) shows results for SHAC with no TD($\lambda$) critic estimation (i.e. no critic but still short horizons). Since a TD($\lambda$) critic is an essential part of SHAC, and not a contribution of our paper, we do not feel the need to include an ablation of it.

---

> > ### Comment · Reviewer_fDX3 · 2024-01-05
> > **Minor comments**
> >
> > Just some minor issues I found in the new version:
> > - Is it supposed to be Columb? Or rather Coulomb?
> > -  “deep learning” is repeated in the last paragraph of page 5.

---

### Author Response · Authors · 2023-12-17
**General response to all reviewers**

We would like to thank the reviewers for their thoughtful and thorough reviews. We appreciate that the reviewers found our approach insightful, theoretically justified, and empirically validated. We have uploaded a revised version incorporating the reviewer's feedback. In the supplementary material, you can also find a pdf highlighting the changes made. Note that it does not highlight changes in figures or algorithms. We address concerns shared by reviewers below:

### 1) Unclear what simulator is used (Reviewers [TQaX](https://openreview.net/forum?id=fnBAaUksGL&noteId=FMQ61s6vOf) and [fDX3](https://openreview.net/forum?id=fnBAaUksGL&noteId=ZjmLcVsMfq))
We acknowledge the oversight in omitting essential details about the simulator used. Our simulations were conducted using the dflex simulator, initially developed for SHAC, the predecessor of our method. We have clarified this by explicitly referencing the dflex simulator in our experiment section and providing additional information in Appendix D. Furthermore, we have open-sourced both the simulator and the environments used in our study. It is important to note that the generalizability of our method extends to other differentiable simulators employing alternative methods of rigid body and contact simulation.

### 2) Inconsistencies with the Heaviside example (Reviewers [TQaX](https://openreview.net/forum?id=fnBAaUksGL&noteId=FMQ61s6vOf) and [fDX3](https://openreview.net/forum?id=fnBAaUksGL&noteId=ZjmLcVsMfq))
The reviewers pointed out issues with the example presented in Section 3, a critical component for grasping the intuition behind our proposed method. We have reworked the example to improve comprehension and ensure mathematical accuracy. Accompanying figures have also been updated, and we have released the code for this illustrative example to enhance understanding. We note that the takeaway does not change despite the change in the Heaviside function.

### 3) Evidence of SHAC getting stuck in local minima (Reviewers [TQaX](https://openreview.net/forum?id=fnBAaUksGL&noteId=FMQ61s6vOf) and [DU72](https://openreview.net/forum?id=fnBAaUksGL&noteId=qt4HZcNOQd))
Reviewers felt that our conclusion from Section 4.1 that SHAC gets stuck in local minima on the Hopper task doesn’t have sufficient evidence. From the results in Figure 7 on the Hopper task, we observe that around 200k steps, SHAC’s performance starts to deteriorate across five different runs. However, AHAC’s reward curve keeps monotonically increasing. Based on this and the biassed first-order gradients, we conjecture that SHAC’s poor performance is due to it getting stuck in local minima. We have added an additional pointer to this in our updated draft.

### 4) Open-source code
Additionally, we provide a link to the code that will later be open-sourced for this paper. It contains both the simulator and our main contribution, AHAC, as well as all baselines used in the paper. https://anonymous.4open.science/r/DiffRL-1D68/README.md

---

> ### Comment · Reviewer_fDX3 · 2024-01-05
> **General response's response**
>
> The Heaviside example is much more readable now, but I think the math is incorrect. In particular, the error function erf is inversely correlated with $\theta$, and it is different to $0$ when $\theta$ is $0$ (assuming $\nu$ is not $0$). These differences indicate that your calculations do not correspond to Figure 2.
>
> I also find your claim about SHAC getting stuck in local minima not supported by the evidence. The reason might be related with alternative issues like exploding gradients or loss of plasticity, which do not necessarily correspond to any local minima.

---

> > ### Author Response · Authors · 2024-01-08
> > **Response**
> >
> > > The Heaviside example is much more readable now, but I think the math is incorrect. In particular, the error function erf is inversely correlated with $\theta$, and it is different to $0$ when $\theta$ is $0$ (assuming $\nu$ is not $0$). These differences indicate that your calculations do not correspond to Figure 2.
> >
> > Thank you for highlighting this. Indeed, our calculations loosely followed Suh et al. 2023 and had errors. We have redone the calculations of true gradients and added them to Appendix A. Furthermore, we also decided to explicitly include the true gradient in Section 3, but instead focus on the empirical bias. With that in mind, we have also replaced Figure 3 to now show the gradient bias instead of the gradient estimate, which we think shows our point better.

---

### Decision · Action_Editor_R8UT · 2024-01-26

**Recommendation:** Reject

**Comment:**

In the end, one reviewer voted weak reject, one reviewer voted weak accept (but acknowledged the weaknesses raised above during the discussion and that many of the ACs suggestions should be address). The third voted accept, but did not participate in the discussion and their review not detailed.

Again, I thank the authors for their engagement in improving the work, it just needs a couple more iterations. Certainly a good candidate for resubmission of a major revision.

**Audience:**

This is mostly ok, but it is also true that the idea of using a simulator as a model requires more justification then given in the current paper:
- what are the applications were we have the model of the world of this type and want to learn optimal policies?
- if the idea is sim2real, the paper should state this and provide some commentary on the current state of sim2real in general (currently it is mentioned once at the very end of the paper)

**Claims And Evidence:**

This paper has made significant progress over the review period. However, in the AE's judgement there are just too many things to fix (some that were already requested that got missed). The iteration has to stop at some point. The major issues related to claims and evidence are as follows:
- the paper shows significant gains compared to SAC and PPO, but uses a reward normalization that makes it difficult to compare to standard well known benchmark results (e.g., https://spinningup.openai.com/en/latest/spinningup/bench.html or other papers)
- furthermore, the papers claim baseline algorithms were tuned for each domain (contrary to standard practice), but importantly revealing high-performance defaults hypers from the literature for these baselines were not used, and without explanation
- the results with SAC and PPO show PPO is often significantly better. This does not match published results and should be explained
- the ablation study compares untuned ablations. This is not incorrect, but the text should be very clear that without additional hyper-parameter tuning the conclusions from such a study are very limited
- the paper mostly reports IQM and CI but in places still reports standard derivation (without explanation) and still makes claims about lower standard derivation without statistical evidence
- IQM is used without justification. Dropping worst and best performance should be justified in text---this involves more than citing Agarwal et al. 2021
- some methods like SVG perform worse than prior work, but this is not explained
- hundreds of parallel actors are used but its unclear how that is reflected on the x-axis of the plots. This could explain SAC's poor performance (contrary to the results reported by Amos et al)---is it getting the same amount of data?
- the simulator used is dflex which has unstated relationships to mujoco and aigym. Makes comparisons with the literature very challenging. How does Hopper used here compare to Hopper-v3?

- as reviewer fDX3 noted, in the end it remains unclear why the new method is better. Not a correctness issue, but would be great to dig deeper.

There remains a few to many issues to address.

**Resubmission Of Major Revision:**

The authors may consider submitting a major revision at a later time.